# Systematic evaluation of blood contamination in nanoparticle-based plasma proteomics

Huanhuan Gao[1,2,3,9], Yuecheng Zhan[4,9], Yuanqi Liu[5,9], Zhiyi Zhu[4], Yuxiu Zheng[4], Liqin Qian[1,2,3], Zhangzhi Xue[1,2,3], Honghan Cheng[1,2,3], Zongxiang Nie[1,2,3], Weigang Ge [ID][4], Senlin Ruan[6], Jiaxu Liu[7], Jikai Zhang[7], Yingying Sun[1,2,3], Lei Zhou[1,2,3], Dongyue Xun[8], Yingrui Wang [ID][1,2,3], Heyun Xu[5✉], Huiwen Miao [ID][5✉], Yi Zhu [ID][1,2,3✉] & Tiannan Guo [ID][1,2,3✉]

## Abstract

**Circulating blood proteomics enables minimally invasive biomarker discovery. Nanoparticle-based circulating plasma proteomics studies have reported varying number of proteins (ca 2000–7000), but it remains unclear whether a higher protein number is more informative. Here, we first develop OmniProt—a silica-nanoparticle workflow optimized through a systematic evaluation of nanoparticle types and protein corona formation parameters. Next, we present an Astral spectral library for 10,109 protein groups. Using the Astral with 60 sample-per-day throughput, OmniProt identifies ca 3000 to 6000 protein groups from human plasma. Platelet/erythrocyte/coagulation-related contamination artificially inflates protein identifications and compromises quantification accuracy in nanoparticle-enriched samples. Through controlled contamination experiments, we identified biomarkers for platelet/erythrocyte/coagulation-related contamination in nanoparticle-based plasma proteomics. We developed open-access software Baize for contamination assessment. We validated the pipeline in 193 patients with CT-indistinct benign nodules or early-stage lung cancers, flagging five contaminated samples. This study reveals that contamination alters protein identification/quantification in nanoparticle-based plasma proteomics and presents Baize software to evaluate it.**

**Keywords** Blood Proteomics; Platelet/Erythrocyte Contamination; Protein Corona; Mass Spectrometry; Nanoparticle
**Subject Category** Proteomics

## Introduction

Circulating blood proteomics has emerged as a pivotal research field for discovering disease-related biomarkers, due to sampling accessibility and the rich molecular insights derived from blood constituents (Cai et al, 2023; Niu et al, 2022; Niu et al, 2025). The largest proteome study of about 600,000 circulating blood samples collected in UK Biobank has been initiated and supported by multiple pharmaceutical companies, as encouraged by the success of a pilot study of about 50,000 blood proteomes measured by Proximity Extension Assay (Dhindsa et al, 2023; Eldjarn et al, 2023; Sun et al, 2023). The SomaLogic slow off-rate aptamer (SOMAmers) (Gold et al, 2010) has also been used for thousands of samples (Ferkingstad et al, 2021; Folkersen et al, 2020). Unlike affinity-based technologies requiring a predefined protein target panel, mass spectrometry (MS)-based proteomics (Guo et al, 2025) enables unbiased discovery without a priori protein selection. However, MS-based analysis suffers from limited proteome coverage due to the wide dynamic range of protein abundances in blood samples. Recent circulating blood proteome studies based on MS (Deutsch et al, 2021; Geyer et al, 2024; Guo et al, 2025) have achieved early successes also in disease biomarker discovery and understanding the pathogenesis of disease onset and progression (Cai et al, 2023; Niu et al, 2022; Niu et al, 2025). The protein groups detected from blood using MS have been increased from about 300 (Cai et al, 2023; Geyer et al, 2016; Niu et al, 2022) to over 1000 using high-abundance protein depletion kits (Shen et al, 2020) and acid-assisted protein depletion (Albrecht et al, 2025; Korff et al, 2025; Viode et al, 2023).

In recent years, multiple nanoparticles (NPs) and MS-based methods further increased the protein identification to multiple thousand (Blume et al, 2020; Liu et al, 2022; Ma et al, 2023; Wang et al, 2024). Selected NPs selectively adsorb circulating plasma proteins through non-covalent interactions (e.g., electrostatic, hydrophobic forces), forming a stratified protein corona

[1]Westlake Center for Intelligent Proteomics, State Key Laboratory of Medical Proteomics, Westlake Laboratory of Life Sciences and Biomedicine, Hangzhou, Zhejiang Province, China. [2]Affiliated Hangzhou First People's Hospital, School of Medicine, Westlake University, Hangzhou, Zhejiang Province, China. [3]Research Center for Industries of the Future, School of Life Sciences, Westlake University, Hangzhou, Zhejiang Province, China. [4]Westlake Omics (Hangzhou) Biotechnology Co., Ltd., Hangzhou, China. [5]The First Affiliated Hospital, Zhejiang University School of Medicine, Hangzhou, China. [6]Department of Clinical Laboratory, Affiliated Hangzhou First People's Hospital, Hangzhou, Zhejiang Province, China. [7]State Key Laboratory of Fine Chemicals, Frontier Science Center for Smart Materials, School of Chemical Engineering, Dalian University of Technology, Dalian, China. [8]College of Chemistry, Nankai University, Tianjin, China. [9]These authors contributed equally: Huanhuan Gao, Yuecheng Zhan, Yuanqi Liu. ✉E-mail: xuheyun@zju.edu.cn; randy_m@zju.edu.cn; zhuyi@westlake.edu.cn; guotiannan@westlake.edu.cn

categorized into hard and soft layers (Mahmoudi et al, 2023). The "soft" corona, initially dominated by high-abundance proteins, is progressively replaced by a "hard" corona of high-affinity proteins through competitive displacement over time, as described by the Vroman effect (Vroman, 1962). The hard corona comprises tightly bound proteins with slow exchange rates, whereas the soft corona consists of loosely associated proteins in dynamic equilibrium with the surrounding environment, conferring distinct stability profiles under physiological conditions (Ke et al, 2017). Beyond the application of diverse NPs in blood proteomics, emerging studies have shown that introducing small molecules to modulate NP protein corona diversity can achieve enhanced enrichment of low-abundance proteins (Pringels et al, 2018; Ashkarran et al, 2024; Tang et al, 2023). Increasing number of NPs have been developed to measure circulating blood proteomes, reporting very different numbers of protein groups, ranging from ca 2000 to ca 7000, even using similar LC and MS instruments and data analysis strategy. The variability of NP-based proteomic depth presents a major concern in this emerging field (Ashkarran et al, 2022).

A meta-analysis of 210 blood proteomics studies (Geyer et al, 2019) corroborates this premise, identifying contamination markers—including platelets, erythrocytes, or coagulation factors—in 54% (113/210) of analyzed studies. This study also reported higher protein identification in platelets and erythrocytes compared to plasma samples. However, systematic investigations into how increasing platelet/erythrocyte contamination levels affect proteomic identification and quantification remain lacking. Furthermore, the impacts of such contamination in NP-enriched plasma proteomics workflows have not been thoroughly analyzed. We hypothesize that the presence of platelet and erythrocyte contaminants in plasma samples introduces variability in proteomic analysis.

To test whether platelet and erythrocyte contamination is the major sources of the observed huge variability of NP-based plasma proteomics experiments, we optimized a NP-based plasma proteome enrichment workflow called OmniProt capable of identifying ca 3000–6000 protein groups per 24-min gradient Astral nDIA (Guzman et al, 2024) analysis. We also generated a comprehensive spectral library of over 10,000 protein groups detected in human plasma using more than 20 types of NPs. Building upon this analytical framework, we systematically evaluated the impact of platelet and erythrocyte contamination using the NP-based plasma proteomic workflow. With varying degrees of contamination, we identified signature protein biomarkers for each type of contamination. Furthermore, we developed an open-access software tool called Baize to evaluate NP-based plasma contamination. Finally, we evaluated the applications of OmniProt and Baize in a cohort of 193 individuals, including 42 cases of pulmonary benign nodules and 114 small malignant nodules where conventional CT imaging cannot reliably distinguish malignant tumor from the benign.

# Results

## Selection and characterization of nanoparticles

To enrich low-abundance plasma proteins, we optimized our sample preparation using multiple silica-based NPs (Fig. 1). Given that NP-mediated protein adsorption is largely non-specific and influenced by

the physicochemical properties of both proteins and NPs (Madathiparambil Visalakshan et al, 2020), we tested a panel of $SiO_2$ NPs with distinct morphologies and sizes (Dataset EV1). This panel included solid spherical NPs (NP23, 500 nm; NP24, 300 nm; NP25, 800 nm; and NP26, 1000 nm), mesoporous NP (NP84, 100–400 nm), hollow mesoporous NPs (NP85, ~400 nm), and hierarchical porous NPs (NP86, 400–900 nm). To evaluate their performance in plasma proteome analysis, pooled plasma samples from lung disease patients were analyzed in triplicate (15 µL per sample). Overall, over 80% of the total proteins identified by these NPs are shared by different NPs (Appendix Fig. S1a). The reproducibility based on biological replicates is high, with median coefficients of variation (CVs) of 1–2% (Appendix Fig. S1b). These data suggest a high degree of technical reproducibility of this workflow. Solid silica NPs showed better performance, yielding 16% more protein groups and 33% more peptides than other types (Fig. 1B,C). Next, we further examined the solid silica microspheres of different sizes (300, 500, 800, and 1000 nm) using Scanning electron microscopy (SEM) before and after incubation with plasma proteins (Fig. 1D). The SEM picture shows a nearly uniform size and monodispersity for each type of solid silica NPs. After incubation with plasma samples, the diameters of all solid silica NPs increased, which is due to the so-called corona, consistent with previously reported findings (Hajipour et al, 2023). With the physical uniformity of NPs established, we then evaluated their performance in enriching low-abundance proteins from plasma samples. Although the numbers of identified proteins and peptides were similar across the different sizes, practical considerations such as the specific surface area and the centrifugation speeds needed to remove the soft corona favor medium-sized particles. Consequently, the 500 nm NPs (NP23) were selected as the optimal configuration for further characterization.

## Establishment of the OmniProt plasma proteomics workflow

NP-based protein enrichment relies on protein corona formation and soft corona washing, both important for capturing low-abundance plasma proteins (Li et al, 2024; Madathiparambil Visalakshan et al, 2020). Using NP23, we optimized key procedures, including plasma dilution, protein corona formation, corona washing and NP storage. 15 µL aliquots of pooled plasma from lung disease patients were analyzed using a 24-min gradient Astral nDIA analysis in triplicate. To optimize plasma diluent, we compared diluent buffers in acidic (pH 3.0), neutral (pH 7.0), and alkaline (pH 11.0) conditions. Results showed that the alkaline diluent led to on average 36% more peptide identifications and 12% more protein identifications compared to the acidic and neutral conditions (Appendix Fig. S2a). We further confirmed the preserved structural integrity of silica NPs and stable protein corona formation under the alkaline conditions using SEM analysis (Fig. 1D). Next, we optimized NP amount and incubation time. We tried different amounts of NPs ranging from 0.3 mg to 2.0 mg, and found that 0.5 mg slightly outperformed the other conditions (Appendix Fig. S2d). We also incubated the NPs with the plasma for varying durations ranging from 30 min to 120 min, and found 30 min was sufficient (Appendix Fig. S2e). Next, we evaluated the NP washing steps for removing soft corona, and found that three consecutive washes resulted in robust proteomics analysis (Appendix Fig. S2b). We also compared different centrifugation speeds, and the results prioritized $7000 \times g$ as the optimal speed (Appendix

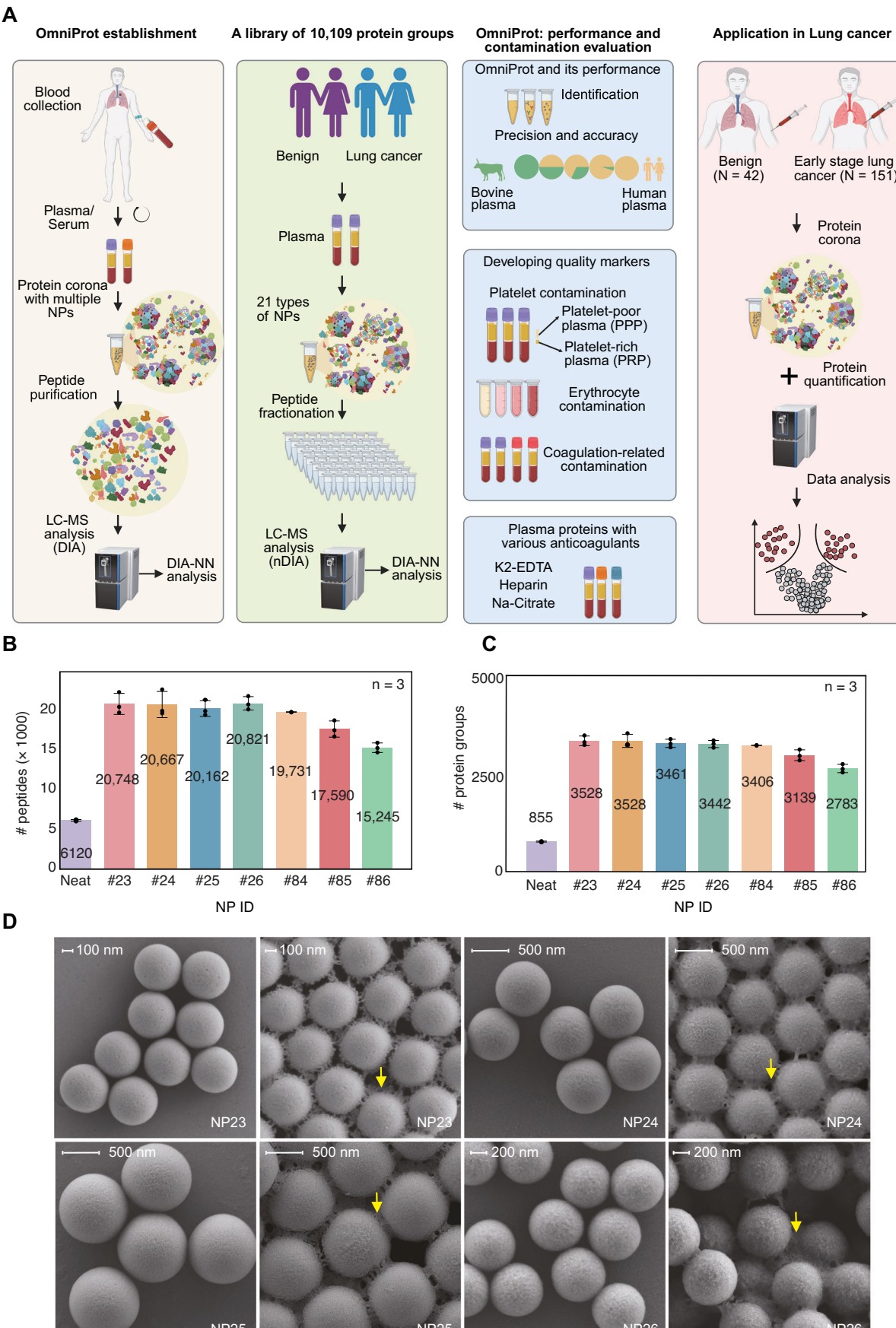

◄

**Figure 1.  Systematic comparison of SiO₂-based nanoparticles for plasma proteomics.**

(A) Schematic overview of the study. Comparison of peptide (B) and protein group (C) identifications across SiO₂-based nanoparticles. (D) Morphological diversity of solid SiO₂-based nanoparticles revealed by Scanning electron microscopy (SEM). Yellow arrows in (D) indicate the protein corona surrounding the nanoparticles. Numerical labels in panels denote cumulative identifications. Data represent mean ± standard error of the mean from biological triplicates. Source data are available online for this figure.

Fig. S2c). Finally, we assessed the thermal stability of NPs by incubating them at 4 °C, 15 °C, and 30 °C for 24 h. The results showed no significant difference in protein identifications, indicating that short-term exposure to room temperature does not affect NP performance (Appendix Fig. S2f). These optimizations established the optimal conditions for protein corona formation, ensuring reliable and reproducible NP-based plasma proteome enrichment. This optimized protocol, based exclusively on NP23, is hereafter referred as OmniProt.

To further ensure the robustness of our optimized protocol, we addressed a potential concern highlighted by recent findings. Sheibani et al (Sheibani et al, 2021) reported that high nanoparticle concentrations during corona formation can induce protein contamination and flocculation artifacts, potentially affecting the accuracy of nanoparticle-based proteomics approaches. To verify whether this phenomenon occurs with our NP23 at the concentrations used in OmniProt, we compared our standard protocol (0.5 mg nanoparticles at 2.78 mg/mL reaction concentration) with a higher concentration condition (0.5 mg nanoparticles at 5.55 mg/mL reaction concentration) using both room temperature transmission electron microscopy (RT-TEM) and cryo-transmission electron microscopy (Cryo-TEM). While low-magnification imaging showed no obvious aggregation under either condition, high-magnification analysis revealed clear evidence of flocculated biomolecules, with more pronounced fine-grained structures observed under the higher concentration condition (Appendix Fig. S3a–f). These results confirm the previous report (Sheibani et al, 2021) that elevated nanoparticle concentrations can indeed induce protein flocculation artifacts.

## Identification of over 10,000 protein groups from NP-enriched plasma proteomics

To construct a comprehensive spectral library for DIA data analysis in plasma proteomics, we utilized 21 chemically distinct NPs to enrich low-abundance proteins in pooled plasma samples from 20 patients with lung disease. The plasma samples were free from significant platelet, erythrocyte and coagulation-related contaminants. Peptide samples from these NP-enriched plasma proteins were fractionated into 30 or 60 fractions via high-pH reversed-phase chromatography (Fig. 2A), followed by nDIA Astral analysis. Altogether, we generated 780 DIA files (Dataset EV2), which were analyzed using DIA-NN. The number of proteins identified in different NP samples ranged from 3325 to 7129 (Fig. 2C). Across all files, we identified 126,661 peptide precursors mapping to 10,109 protein groups (Fig. 2D). The Human Plasma Atlas (Ou et al, 2011) identified 4066 protein groups, of which 3819 protein groups (94%) are covered by our data resource (Fig. 2D). We then analyzed the proteins enriched by each NP of various charge, functional group, matrix, hydrophobicity and the reaction class (Fig. 2B). The number of protein groups identified from the 21 NPs varied, with

the lowest number observed for polystyrene NPs and the highest for silica-based NPs. Bioinformatics analysis of the 10,109 protein groups using Gene Ontology Biological Process (Ashburner et al, 2000; Gene Ontology et al, 2023) and KEGG (Kanehisa et al, 2021) databases showed predominant functional associations with signal transduction, protein phosphorylation, proteolytic regulation, apoptotic mechanisms, cellular adhesion and innate immunity (Fig. 2E). KEGG pathway analysis identified 207 significantly enriched pathways (FDR-adjusted $p < 0.05$), with the top decile comprising metabolic cascades, neurodegenerative disorders and oncogenic pathways (Fig. 2F), suggesting that this resource covers diverse physiological and pathological processes.

## Performance of OmniProt in measuring low-abundance plasma proteins

Next, we employed this spectral library to perform a comparative analysis of OmniProt against both a commercial High-Selected top-14 abundant protein depletion kit (Top14) and direct digestion of plasma samples (neat). The evaluation utilized platelet-poor plasma (PPP) samples from 22 patients with lung disease, followed by nDIA in an Astral mass spectrometer. Our results showed that the OmniProt method identified a median of 2771 protein groups and 15,816 peptides, yielding 2.6- and 5.7-fold more peptides (Fig. 2G) and 1.7- and 4.8-fold more protein groups (Fig. 2H) compared to the Top14 and the neat condition, respectively. Compared to neat and Top14, OmniProt identified a greater number of lower-abundance protein groups, with the highest concentration observed in the log intensity range of 3.2 to 4.5 (Fig. 2I). The above results initially showed the improved sensitivity of the OmniProt workflow in detecting low-abundance plasma proteins. However, the overall identification numbers remained lower than expected from our subsequent experiments (described below). This lower identification was mainly caused by plasma quality issues from long-term frozen storage and repeated freeze-thaw cycles. In later experiments with freshly collected plasma samples, the OmniProt method successfully detected 2500–4200 protein groups per plasma sample (Fig. 7B).

Next, to evaluate the precision and accuracy of OmniProt-based proteome analysis, we designed a mixed-species benchmarking experiment (Huang et al, 2025). We spiked bovine plasma samples into a pool of human plasma samples at increasing ratios (0% to 100% v/v). The mixed plasma samples were analyzed by nDIA on an Astral either directly or after OmniProt enrichment for low-abundance proteins. Based on the experimental design shown in Fig. 3A, we defined seven bovine concentration groups (A–G) that were used consistently in all subsequent analyses (Fig. 3B–E). The OmniProt samples led to identification of 36,977 peptide precursors derived from 4950 protein groups across all dilution conditions, with an 8-fold increase in protein identifications compared to the neat samples (Fig. 3B,C). The OmniProt samples

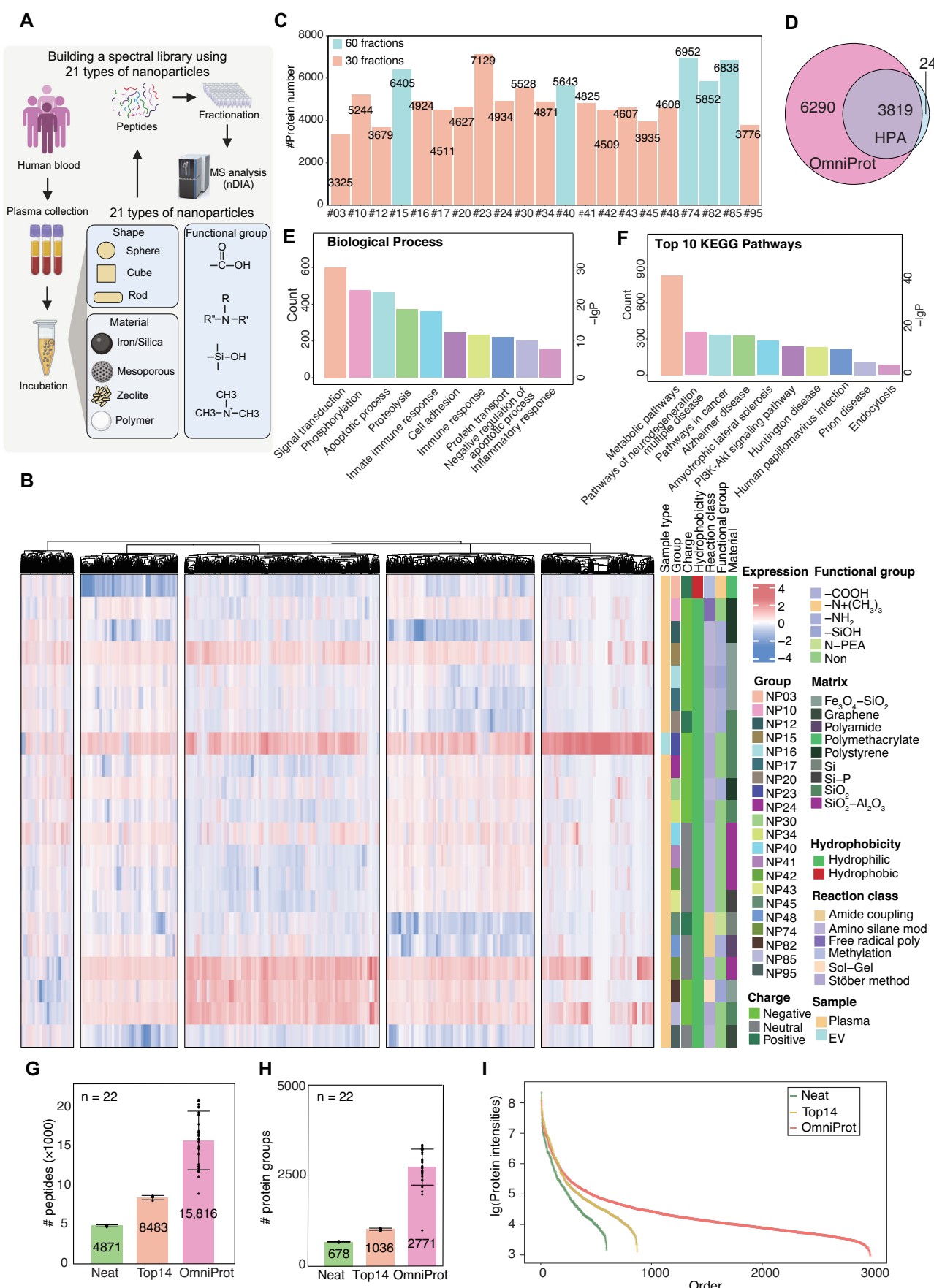

Figure 2.   Nanoparticle-enhanced plasma spectral library construction and performance comparison.

(A) Schematic overview of nanoparticle-driven spectral library generation. (B) Heatmap showing 10,109 protein groups identified across 21 nanoparticle types. (C) Number of proteins identified by each of the 21 NP types. (D) Comparative analysis with the Human Plasma Atlas (HPA). Top 10 enriched biological process (E) and KEGG pathways (F) ranked by GO analysis using DAVID. Identified peptides (G), protein groups (H) and protein abundance ranking (I) in plasma samples processed with neat, Top14 kit, and OmniProt methods. Data represent mean ± standard error of the mean from 22 patients. Color scale in (B) represents Z-score (standard deviations from mean), linear scale, range: −4 to +4. Source data are available online for this figure.

also showed comparable quantitative reproducibility to the neat samples across all spike-in ratios, with biological triplicates exhibiting median CVs < 20% (Fig. 3D). To validate quantification reliability, we confirmed that coefficient of variation correlated inversely with protein abundance under both neat and OmniProt conditions. Importantly, median CVs remained below 20% across all groups (A–G as defined in Fig. 3A), meeting standard requirements for quantitative proteomics. Furthermore, CV does indeed correlate inversely with protein abundance under both neat and OmniProt conditions—higher abundance proteins exhibit lower CVs, while lower abundance proteins show higher CVs (Appendix Fig. S4).

To assess quantitative accuracy, we focused exclusively on bovine-specific peptide precursors to eliminate interference from shared peptide precursors between the two species. This methodology systematically employs a broad bovine concentration gradient series (groups A–G) to enable rigorous benchmarking of experimental fold-change measurements against theoretical expectations under six predefined spiked-in ratio conditions. We calculated fold-change ratios of bovine-specific peptide precursors between six predefined group pairs, using the known spiked-in ratios as ground truth. Results indicated that quantitative accuracy in multi-species samples remained generally consistent across dilution levels for both the neat and the OmniProt samples (Fig. 3E). Overall, these results show better performance of the NP workflow, yielding more proteins than the neat plasma, and high degree of quantitative precision and accuracy in NP-based plasma proteomics.

## Variability of protein identification in different plasma samples

During the optimization and application of the OmniProt workflow, we observed that this protocol, when applied to different plasma samples, led to a large variation in the number of protein identifications, ranging from ca 3000 to 7000 (Appendix Fig. S5a–d). Then we inspected the data, and found that the samples with high number of protein identifications contained abnormally high levels of platelet-derived and/or erythrocyte-derived proteins, suggesting potential contamination of platelets and erythrocytes, which was subsequently thoroughly investigated as described below.

## Platelet contamination and markers

To systematically assess platelet contamination in NP workflows, we collected platelet-rich plasma (PRP) and the PPP samples from two donors (one male and one female), pooled the respective PRP and PPP samples from each donor, and mixed them at seven predefined volumetric ratios (Fig. 4A). The mixed samples were processed using the neat and the OmniProt workflows in triplicate, followed by nDIA analysis. Our results showed that platelet spike-in increased peptide precursor and protein group identifications in both neat and OmniProt samples (Fig. 4B,C). The OmniProt workflow identified 4580 protein groups on average in the PRP samples versus 2492 in the PPP samples. Next, we examined whether platelet contamination alters the concentration of circulating plasma proteins after NP enrichment. Across all samples with varying degrees of platelet contamination, a total of 4842 protein groups were identified. The median Spearman correlation between the PRP/PPP ratio and protein intensities was 0.661 (Appendix Fig. S6a). Our data show that while some proteins showed positive correlations with increasing platelet concentrations, others exhibited no correlation or even negative correlations with platelet levels. Then, we identified 2432 proteins in pure PPP samples with no significant association with platelet levels (correlation coefficient < 0.7). These proteins are regarded as platelet-independent circulating proteins to a certain extent. Our data showed that the abundances of these proteins in the PPP samples exhibited relatively weak correlations (r = 0.54–0.73) with those in plasma samples with various degrees of platelet contamination (Appendix Fig. S6b). This indicates that NP-based enrichment moderately alters these platelet-independent circulating proteins, thereby compromising the quantitative accuracy of circulating plasma proteome analysis. Thus, it is non-trivial to assess platelet contamination in NP-based plasma proteome analysis.

Geyer et al (Geyer et al, 2019) have identified 30 proteins as biomarkers for platelet contamination in neat plasma proteome analysis. Next, we evaluated whether these protein markers could be applied to evaluate platelet contamination in the above-mentioned neat and OmniProt samples. As expected, the neat samples showed strong correlation (median r = 0.95) across 30 protein markers, confirming the findings from Geyer et al (Geyer et al, 2019). However, the OmniProt samples showed relatively weaker correlation (median r = 0.75) as shown in Appendix Fig. S6c,d. The protein GSN even showed a negative correlation with increasing platelet contamination. Therefore, there is a need to further define a customized contamination list for NP-based plasma proteome experiments.

To identify protein biomarkers for monitoring platelet contamination, we first filtered out all the protein groups with missing values exceeding 50% across all samples, leading to 4404 consistently detected protein groups, which were subsequently clustered into eight distinct groups using Mfuzz. Mfuzz clustering revealed two groups with different abundance trends. The proteins in the first group gradually increased with platelet concentrations until plateau, while the proteins in the other group declined as platelet levels increased (Appendix Fig. S7a). We then identified the top 100 proteins with highest correlation with increased platelet

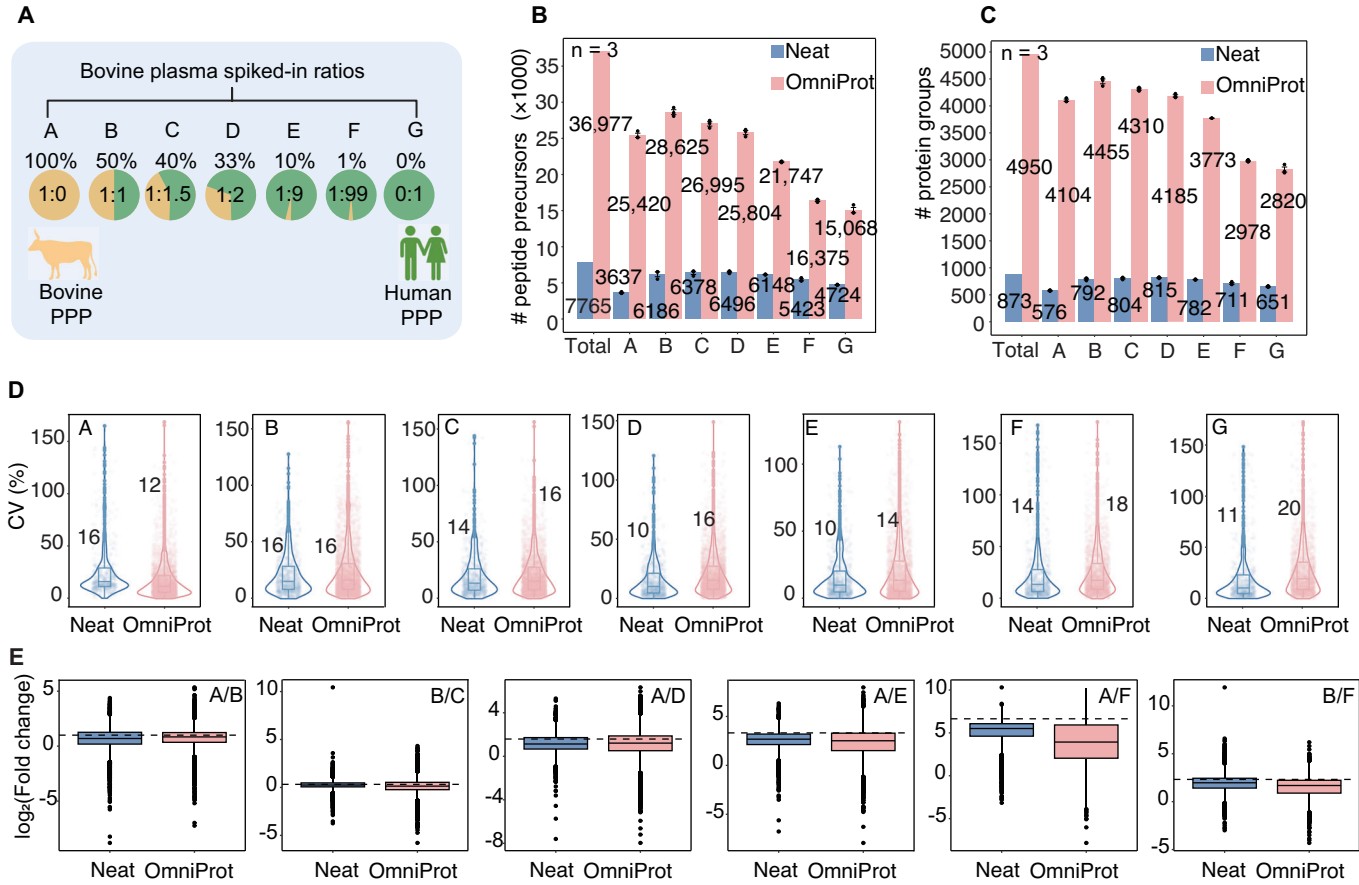

**Figure 3. Benchmarking of the OmniProt workflow.**

(A) Experimental design for spiking the bovine plasma proteome into human plasma across seven concentration groups (A–G). Number of identified peptide precursors (B) and protein groups (C) in bovine-human plasma mixtures. Data represent mean ± standard error of the mean from biological triplicate analyses. (D) Inter-species coefficients of variation across spiked-in ratios. (E) Experimental fold change distributions of bovine-specific peptide precursors versus theoretical expectations (dashed lines) for six pairwise group comparisons. Each panel shows comparisons between two groups (e.g., A vs B, B vs C) with blue boxplots representing neat samples and pink boxplots representing OmniProt samples. The box in each group extends from the first to the third quartile, with a horizontal line indicating the median value. Whiskers extend from the box to represent the range of data within 1.5 times the interquartile range. Source data are available online for this figure.

concentrations. From this subset, the top 30 proteins exhibiting highest abundance levels were retained as candidate biomarkers for platelet contamination. The protein expression of these 30 biomarkers in plasma samples with increasing platelet contamination is provided, with Spearman correlation coefficients ≥0.92 (Appendix Fig. S7c), confirming their correlation with platelet contamination. These proteins exhibit relatively high abundance in the NP-enriched samples (Fig. 4D). The Spearman correlation of our markers showed stronger consistency (median r = 0.95, range r = 0.92–0.98) (Fig. 4E). The selected platelet-associated proteins, including ADP/ATP translocase 2 (SLC25A5) (Woulfe et al, 2001), platelet factor 4 variant 1 (PF4V1) (Pilatova et al, 2013), thromboxane A synthase 1 (TBXAS1) and integrin beta-1 (ITGB1), exhibit key roles in platelet activation and aggregation pathways, substantiating their value as contamination biomarkers. Pathway enrichment analysis revealed that these 30 proteins are primarily associated with mitochondrial protein degradation, mitochondrial calcium ion transport, eicosanoid signaling, sirtuin signaling pathway and cardiac hypertrophy signaling (Enhanced) (Appendix Fig. S7b).

To further validate these 30 biomarkers, pure platelet and PPP samples from 12 patients were procured and pooled into four groups (three patients per group). These platelet samples were subjected to a ten-step serial dilution with the PPP samples. We further performed blood cell counting to confirm the degree of platelet contamination. These samples were analyzed with the OmniProt workflow (Fig. 4F, Dataset EV3). The results confirmed that a higher degree of platelet contamination led to an increased number of peptide precursor and protein group identifications (Appendix Fig. S8a–h). We also confirmed that the abundance levels of the 30 platelet markers reliably reflect the degree of platelet contamination (Fig. 4G).

Next, we developed a computer program called Baize (https://www.guomics.com/software/Baize) to evaluate platelet contamination by calculating the ratio of summed signals from the 30 platelet markers to the total protein intensity. The contamination index remains relatively stable at lower platelet counts (0–$10^5$ counts), while exhibiting a marked upward trend beyond $10^6$ (Fig. 4H). A Spline regression analysis was performed on this elevated portion, with the identified change point serving as our threshold criterion

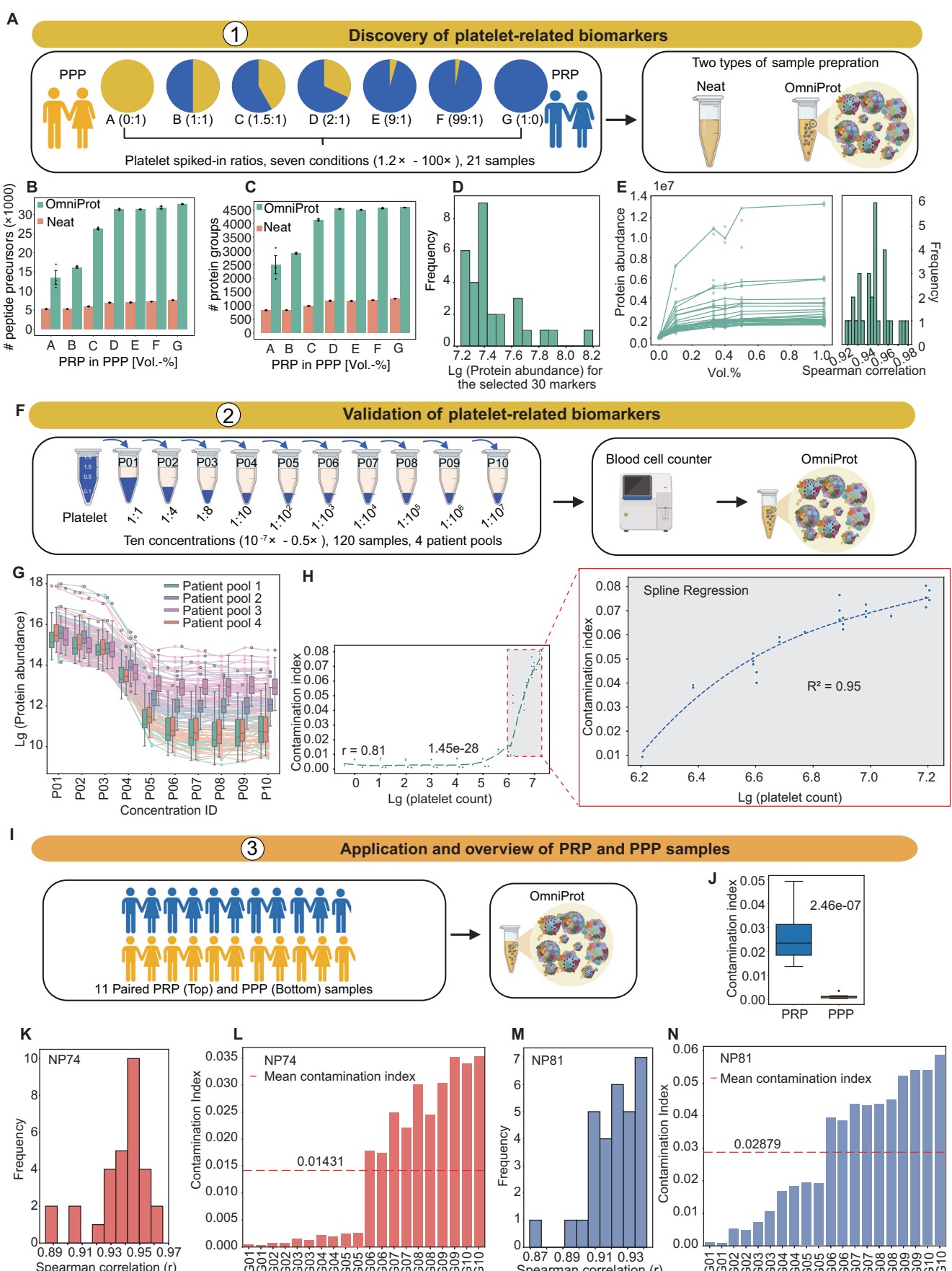

**Figure 4. Quality marker panel for platelet contamination.**

(A) Experimental workflow for platelet-related biomarker discovery. Number of identified peptide precursors (B) and protein groups (C) in discovery dataset. (D) Abundance distribution profiles of 30 selected protein biomarkers. (E) Spearman correlation analysis of platelet-related biomarkers in discovery dataset. (F) Validation workflow of platelet-related biomarkers. (G) Logged protein intensities of 30 platelet-related biomarkers across the dilution series. (H) Spline regression analysis between platelet count and contamination index. (I) Application scheme for platelet contamination assessment. (J) Contamination index calculation in PRP and PPP samples. The Spearman correlation of the 30 platelet-related biomarkers in NP74-processed samples (K) and NP81-processed samples (L) across platelet contamination levels. Contamination index profiles of NP74-processed samples (M) and NP81-processed samples (N) with graduated platelet contamination. G01-G10 represent plasma samples with graduated platelet contamination prepared by mixing platelet-rich plasma and platelet-poor plasma sample at defined ratios (G01: lowest contamination; G10: highest contamination). PRP: platelet-rich plasma; PPP: platelet-poor plasma. Data represent mean ± standard error of the mean from biological triplicate analyses. The box in each group spans from the first to the third quartile, with a horizontal line indicating the median value. The whiskers extend from the box to represent the range of data within 1.5 times the interquartile range. Source data are available online for this figure.

for detecting the onset of significant platelet-mediated contamination. The model achieved an $R^2$ of 0.95, showing strong correlation for systematic platelet contamination assessment in NP-based plasma proteomes. When samples exceed this threshold, they are flagged as potentially compromised by platelet contamination.

The platelet contamination algorithm was further applied to paired PRP and PPP samples obtained from 11 patients with lung cancer (Fig. 4I). The results showed that the platelet contamination index in the PPP samples remained close to zero, while that in the PRP samples ranged from 0.15 to 0.48 (median: 0.22, Fig. 4J). Indeed, the abundance of these 30 proteins differed between the PPP and PRP samples (Appendix Fig. S9). The platelet contamination also influenced the proteome profiling. The neat samples led to identification of 955 protein groups in the PPP samples and 1373 in the PRP samples, while the OmniProt method improved the protein group identification to 2701 in the PPP samples and 4328 in the PRP samples. 94.5% of the proteins identified in the neat samples were also identified by the OmniProt method (Appendix Fig. S10a,b). We performed hierarchical clustering analysis on proteins identified in both OmniProt- and neat-processed PPP and PRP samples. Our data showed that compared to the neat samples, the proteins uniquely identified in the OmniProt samples are primarily distributed in the following lung cell types (Appendix Fig. S10c): Travaglini lung trem2 dendritic cell, Descartes fetal lung myeloid cells, Travaglini lung platelet megakaryocyte cell, and Travaglini lung proximal basal cell. In the OmniProt-processed samples, the PRP samples exhibited elevated levels of proteins linked to lung cell lineages compared to the PPP samples, specifically Travaglini lung platelet megakaryocyte cell and Travaglini lung trem2 dendritic cell.

Finally, we asked whether these biomarkers could be used for the samples prepared by other types of nanoparticles. We selected two frequently reported nanoparticle types: Zeolite NaY (NP74) (Li et al, 2024) and silanol-functionalized $Fe_3O_4$ (NP81) (Blume et al, 2020). We collected the blood sample from a donor and prepared PPP and PRP plasma samples. 100 µL of PPP and PRP plasma samples were pooled and subjected to a ten-step serial dilution. The diluted plasma samples were processed using the two nanoparticles, namely NP74 and NP81, followed by nDIA analysis. Our results confirmed that increasing platelet concentration elevated peptide precursor and protein group identifications for both NP74- and NP81-processed plasma samples, reaching over 6000 protein groups per injection (Appendix Fig. S10d–g). The 30 platelet-related biomarkers showed a high degree of correlation with increasing platelet contamination in NP74- (median r = 0.95, range r = 0.89–0.96) and NP81-processed samples (median r: 0.93,

range r = 0.87–0.94). Thus, our results further support the applicability of the platelet-related biomarkers identified in this study for evaluating platelet contamination in different NPs (Fig. 4K–N). Notably, the samples with identical platelet contamination levels showed different degrees of contamination between NP74- and NP81-processed samples, ranging from 0.000 to 0.055 for NP81 and 0.000 to 0.035 for NP74 (Fig. 4K–N). These findings indicate that the degree of platelet contamination cannot be directly compared between samples processed through different NP workflows. Therefore, when using alternative nanoparticles, researchers should refer to the contamination index generated by the Baize software. This recommendation has been integrated into the latest Baize V3.0. Platelet contamination is a common issue in existing sample cohorts. Baize software provides a contamination index to identify outlier samples in plasma cohorts.

Building on this flexible assessment approach, we recommend maintaining consistent platelet contamination levels within cohorts, instead of requiring the exclusive use of PPP samples. This strategy is particularly relevant for platelet-related disease research, where PRP samples are directly employed in sports medicine, tissue repair (Alsousou et al, 2012), hair loss treatment, and cardiac applications (Marques et al, 2014). However, for research projects not specifically focused on platelet biology, we strongly recommend using PPP samples, as this approach enables deeper investigation of low-abundance plasma proteins in disease contexts while avoiding interference from high-abundance platelet proteins. To optimize PPP collection, we systematically compared different centrifugation conditions. First, we collected PRP samples using centrifugation at $200 \times g$ for 15 min as a comparison group (group A in Appendix Fig. S11). Subsequently, we collected PPP samples through different centrifugation methods (group B–E in Appendix Fig. S11), including single centrifugation at $2000 \times g$ for 15 min, double centrifugation at $2000 \times g$ for 15 min, single centrifugation at $4000 \times g$ for 15 min, and single centrifugation at $3000 \times g$ for 30 min. Based on protein identifications and contamination index values, the latter three conditions (group C-E in Appendix Fig. S11a,b) showed no significant differences. We found that the Spearman correlation of commonly identified proteins between PPP samples obtained using the latter three centrifugation conditions was also high, all above 0.8 (Appendix Fig. S11c). Therefore, we recommend the more efficient $4000 \times g$ for 15 min protocol for PPP sample preparation due to its shorter processing time.

Once contamination is detected, several remediation options are available. While sample removal remains the primary consideration, alternative approaches can be employed when exclusion is not

feasible. High-speed centrifugation (30 min at $3000 \times g$) has been recently proposed to partially reduce contamination impact (Korff et al, 2025).

## Erythrocyte contamination and markers

Hemolysis represents another important preanalytical variable in plasma proteomics. To systematically assess erythrocyte contamination in NP-based plasma experiments, we applied a similar strategy to the platelet contamination algorithm for systematic analysis. We collected pure erythrocytes and PPP plasma samples from two donors (one male and one female) and pooled the respective erythrocyte and PPP plasma samples from each donor separately. Then, we mixed the erythrocytes and PPP plasma samples with six pre-defined volumetric ratios (Fig. 5A). The mixed samples were processed with the neat plasma method and the OmniProt workflow in triplicate, followed by nDIA analysis. Our results showed that in both neat and OmniProt samples, erythrocyte spike-in led to increased identification of peptide precursors and protein groups, except for the 100% erythrocyte samples (Fig. 5B,C). Next, we investigated whether erythrocyte contamination affects circulating plasma protein concentrations following OmniProt enrichment. We identified 986 proteins in the PPP samples showing no significant association with erythrocyte levels (Spearman correlation coefficient $r < 0.7$). Likewise, these proteins are regarded as erythrocyte-independent circulating proteins to a certain extent. Our data revealed weak correlations ($r = 0.59$–$0.93$) between the abundance of these proteins in the PPP and the erythrocyte-contaminated plasma samples (Appendix Fig. S12b). These observations show that NP-based enrichment perturbs even erythrocyte-independent circulating proteins, ultimately compromising the quantitative accuracy of the circulating plasma proteome analysis. To further evaluate the abundance distribution of erythrocyte-independent circulating proteins and circulating plasma proteins, we categorized protein distributions in samples with various erythrocyte contamination levels through Mfuzz clustering analysis, revealing four clusters of proteins elevated with increasing RBC contamination, while another four clusters showed reverse pattern (Appendix Fig. S12b).

Geyer et al (Geyer et al, 2019) have identified 30 proteins as biomarkers for erythrocyte contamination in neat plasma proteome analysis. Here, we evaluated whether these protein markers could be applied to evaluate erythrocyte contamination in the above-mentioned neat and OmniProt samples. As expected, the neat samples showed strong correlation ($r = 0.98$–$1.00$) across the 30 protein markers, confirming the findings from Geyer et al (Geyer et al, 2019). However, the OmniProt samples showed relatively weaker correlation ($r = 0.75$–$0.98$), as shown in Appendix Fig. S6e,f. To identify protein biomarkers for monitoring erythrocyte contamination, we first selected 100 proteins showing the strongest correlation with erythrocyte levels, and then selected the top 30 candidates based on protein intensity for erythrocyte contamination assessment. The 30 candidate proteins exhibited robust correlation (Fig. 5D,E; Appendix Fig. S13a,b). These proteins were rarely detected in the plasma samples without erythrocyte contamination but showed higher intensities in the 10% erythrocyte samples (Appendix Fig. S14a). Pathway enrichment analysis revealed that these markers are primarily associated with erythrocyte gas exchange (oxygen uptake/carbon dioxide release

and vice versa), neutrophil degranulation, JAK-STAT signaling-mediated gene/protein expression following IL-12 stimulation, and COPI-mediated anterograde transport (Appendix Fig. S13C). We then evaluated the analytical performance of our 30 identified biomarkers in those samples. The Spearman correlation of our markers showed stronger consistency (median $r = 0.987$) (Fig. 5E), including erythrocyte-specific markers with distinct biological roles. Hemoglobin subunits (Ou et al, 2011) (HBD, HBB, HBA1), essential for oxygen transport, showed strong RBC specificity. The membrane-stabilizing spectrins (Chen et al, 2023) (SPTA1, SPTB) maintain RBC structural integrity, while SLC4A1 (Jennings, 2021) regulates gas exchange through anion transport.

To further validate these biomarkers, the erythrocyte and the PPP samples from six patients (Dataset EV3) were collected and pooled into two groups (three patients per group). These erythrocyte samples were subjected to a ten-step serial dilution with the PPP samples (Fig. 5F). We further performed blood cell counting to confirm the degree of erythrocyte contamination. These samples were analyzed with the OmniProt workflow. The results confirmed that a higher degree of erythrocyte contamination led to an increased number of peptide precursor and protein group identifications (Appendix Fig. S14a). We also confirmed that the abundance levels of the 30 erythrocyte markers reliably reflect the degree of erythrocyte contamination (Fig. 5G). Next, we developed erythrocyte contamination indices following established methodologies previously applied to platelet contamination assessment and implemented in the Baize software. This metric exhibited strong robustness, showing a strong linear correlation ($R^2 = 0.94$) with cell counts obtained by hematocytometer (Fig. 5H).

## Variability introduced by coagulation

Improper blood collection represents a major yet often overlooked source of variability in plasma proteomics. Proper collection of plasma samples requires immediate gentle inversion after blood withdrawal to ensure uniform mixing of anticoagulants, followed by centrifugation to yield cell-free plasma. Delays in this process risk partial coagulation, creating hybrid plasma-serum matrices that distort biomarker quantification. Plasma samples are typically collected using anticoagulant-containing tubes: EDTA (purple-top), sodium citrate (blue-top), or lithium/sodium heparin (green-top). All these tubes are used in various clinical applications; however, no systematic study has investigated their impact in NP-based plasma proteomics.

To evaluate the impact of anticoagulants on plasma proteomics, we procured plasma samples from 10 donors (5 males and 5 females, Table EV1). Each donor contributed triplicate blood samples using the three kinds of anticoagulation tubes. Both neat plasma and OmniProt-processed plasma samples were analyzed by nDIA MS. Our results showed that the plasma samples processed via neat and OmniProt showed no marked differences in the total numbers of peptide and protein identifications across the three tube types. The neat samples led to the identification of 975–1049 protein groups, while the OmniProt samples yielded 2643–3507 protein groups (Fig. 6A). However, only 59.2% of the proteins identified in the OmniProt samples are shared across all the tubes (Fig. 6B), suggesting substantial proteome variability of plasma samples collected in different tubes. Compared to the neat samples, the OmniProt samples exhibited reduced protein signals associated

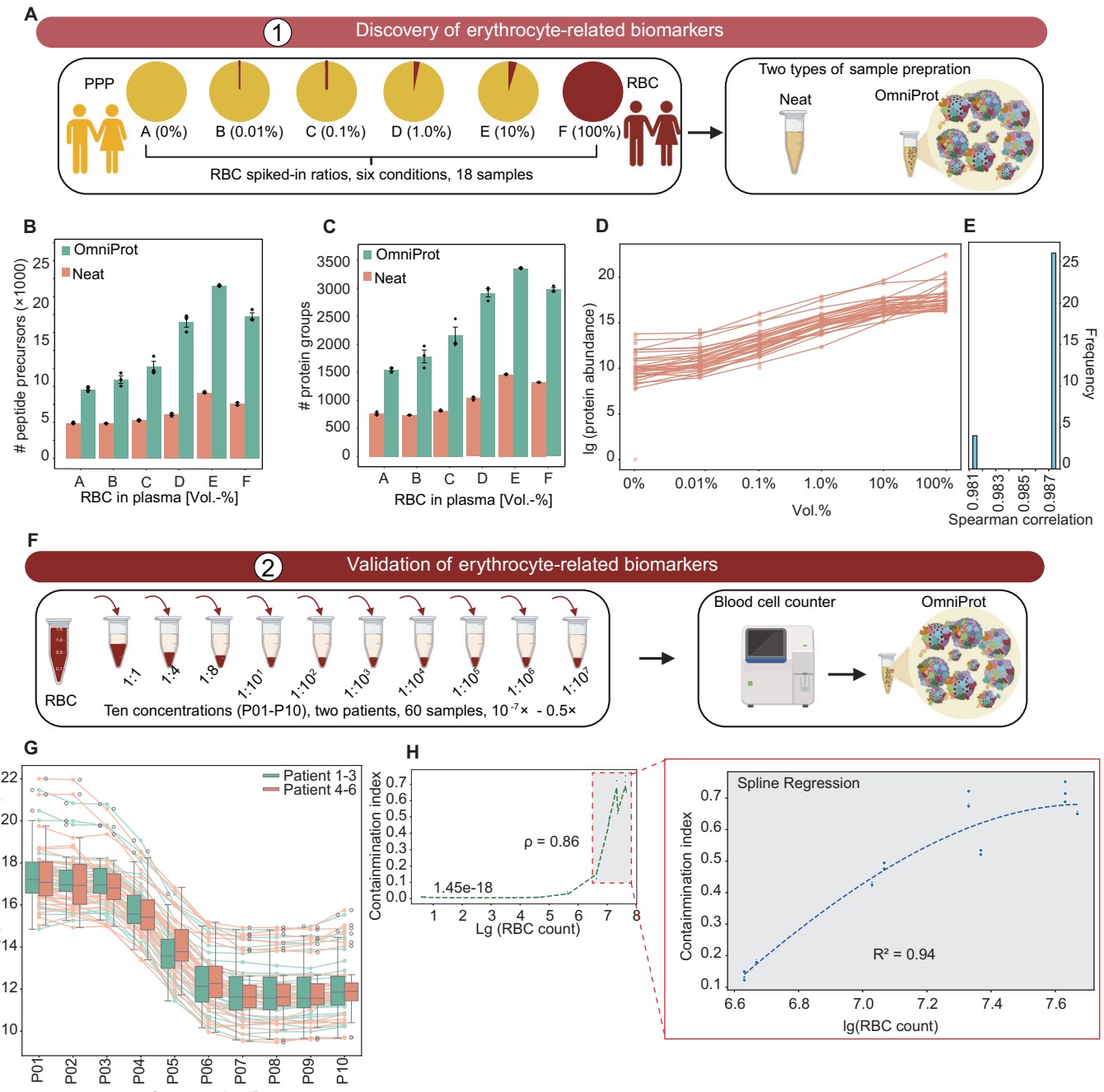

**Figure 5. Quality marker panel for erythrocyte contamination.**

(A) Discovery workflow for erythrocyte-related markers. Number of identified peptide precursors (B) and protein groups (C) in the discovery dataset. (D) Spearman correlation analysis of 30 erythrocyte-related biomarkers in the discovery dataset. (E) Abundance distribution profiles of erythrocyte-associated markers. (F) Validation workflow for erythrocyte contamination markers. (G) Logged protein intensities of erythrocyte markers across dilution series vs. spike-in proportion. (H) Spline regression analysis of erythrocyte count and contamination index. PPP: platelet-poor plasma; RBC: erythrocyte. Data represent mean ± standard error of the mean from biological triplicate analyses. The box in each group spans from the first to the third quartile, with a horizontal line indicating the median value. The whiskers extend from the box to represent the range of data within 1.5 times the interquartile range. Source data are available online for this figure.

with estrogen receptor signaling, mitochondrial dysfunction and related protein degradation sirtuin pathways. Conversely, enhanced enrichment was observed for proteins involved in ROBO receptor signaling, ribosomal quality control, coronavirus replication machinery, eukaryotic translation initiation, B cell developmental

processes, FcRIIB and PI3K signaling in B lymphocytes, as well as IL-15 and p70S6K pathways (Fig. 6C). OmniProt-processed EDTA and heparin plasma samples exhibited greater similarity in protein enrichment profiles compared to the other tube types, with better enrichment of proteins associated with B cell development, FcRIIB

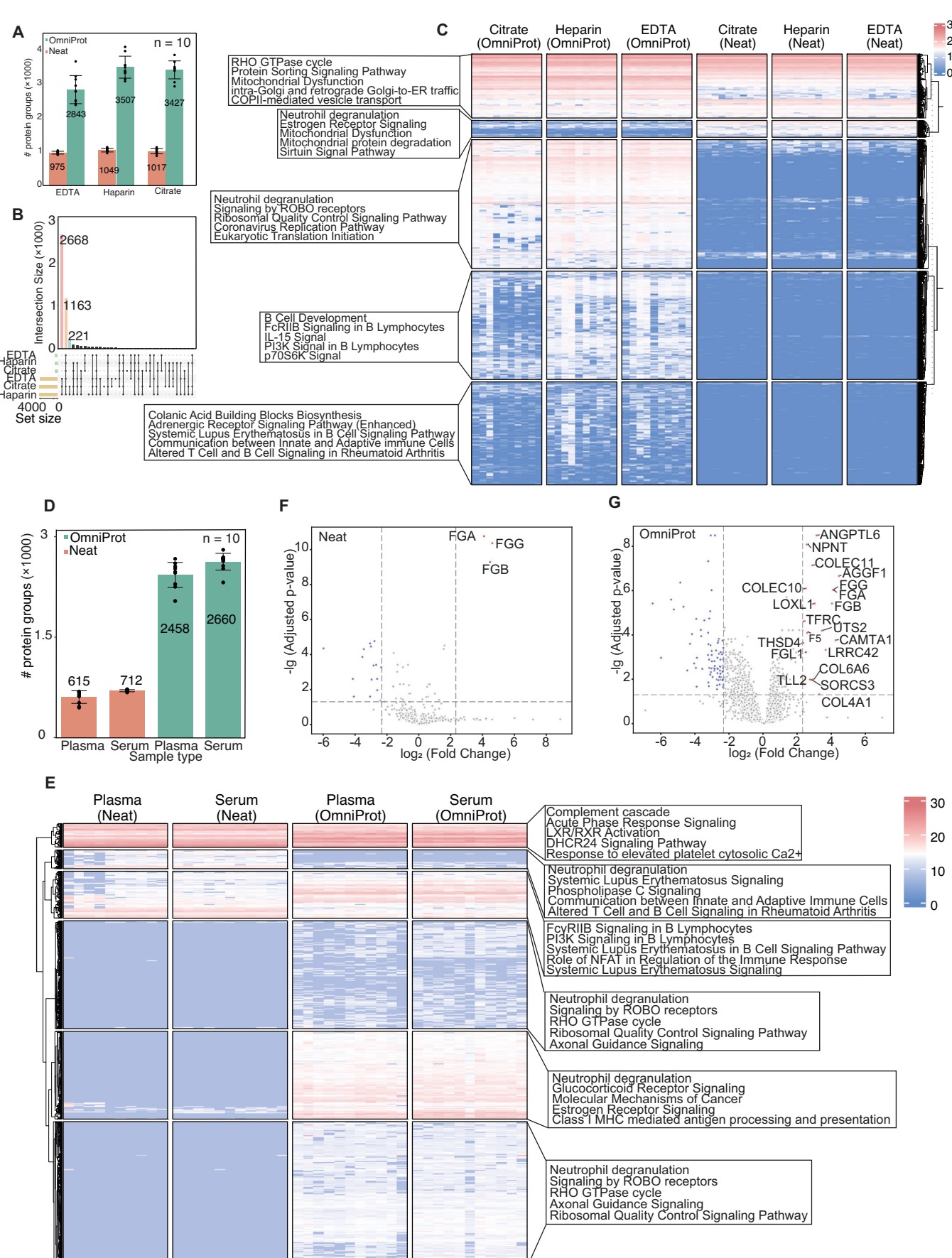

◄ **Figure 6. Performance evaluation of OmniProt in different blood collection tubes.**

(A) Protein group identification yields in the three blood collection tube types. (B) Upset plot of protein overlap across the three blood collection tube types. Data represent mean ± standard error of the mean from 10 patients. (C) Heatmap visualization of tube-specific protein profiles. (D) Protein identifications from 10 paired plasma and serum samples. (E) Heatmap overview of paired plasma/serum proteomes. The color scale represents $\log_2$-transformed protein abundances, log scale, range: 0 to 30. Volcano plots of plasma vs. serum samples processed with neat (F) and OmniProt (G) using paired Student's t test. The colored dots represent those with B–H adjusted $p$-value less than 0.05 and fold change larger than 5. Source data are available online for this figure.

and PI3K signaling in B lymphocytes, IL-15 signaling and p70S6K pathway activation.

We also evaluated the impact of partial coagulation which results in hybrid plasma-serum matrices. Paired plasma and serum samples were collected from the aforementioned 10 donors using EDTA-coated and silica-coated blood collection tubes, respectively. Each donor provided two plasma samples analyzed with neat and the OmniProt methods. The OmniProt proteomic profiling identified 3041 protein groups in the plasma and 3066 in the serum samples, surpassing the neat plasma and serum samples (816 and 796 protein groups, respectively) (Fig. 6D). Heatmap analysis showed that OmniProt-exclusive proteins predominantly mapped to neutrophil degranulation pathways, systemic lupus erythematosus signaling cascades, and ROBO receptor-mediated signaling networks (Fig. 6E). To prioritize coagulation-related proteins, we selected the proteins through the following criteria, requiring a minimum fold-change of 5 and statistical significance (adjusted $p$-value < 0.05) in both processing methods (Fig. 6F,G). These differentially expressed proteins contain FGA, FGG, and FGB which are markedly upregulated in the neat samples, consistent with prior literature (Geyer et al, 2019). The OmniProt samples additionally highlighted 17 known coagulation-related proteins including AGGF1 and TFRC. Finally, we developed a method to evaluate coagulation-related contamination based on these 20 proteins and implemented it in Baize.

## Baize software for evaluating contamination in NP-based plasma proteomics

Here, we introduce Baize, a web-based software tool for rapid evaluation of sample contamination in three major dimensions: platelet contamination, erythrocyte lysis, and residual coagulation protein carryover. The algorithm calculates cell-type-specific contamination index by normalizing the summed intensity of marker proteins against the total intensity of plasma proteome (Contamination Index = Marker Protein Intensities / Σ All Plasma Protein Intensities). Users can submit a protein matrix through a streamlined single-click analysis workflow, and then Baize will output an evaluation of contamination for every plasma/serum sample. This contamination assessment enables researchers to identify contamination issues during sample collection and systematically evaluate contamination patterns across sample sets for quality control and biomarker discovery optimization. The latest version of Baize is freely accessible at https://www.guomics.com/software/Baize.

## Application in a lung cancer cohort

To evaluate the application of OmniProt and Baize, we procured samples from 193 individuals including 42 with benign nodules,

and 151 with early-stage malignancies. In these cases, conventional CT imaging often produces inconclusive results, necessitating surgical resection for definitive diagnosis. We also included six patients with advanced-stage lung cancers as positive controls. Detailed clinical characteristics of the cohort are summarized in Dataset EV4 and Fig. 7A. Plasma specimens were collected as PPP and processed in seven randomized batches using the OmniProt workflow followed by nDIA analysis. We also included seven randomly selected biological replicates and seven randomly selected technical replicates to assess reproducibility. The proteomics analysis led to the identification of 4413 protein groups (Fig. 7B) with stable CVs observed across both biological and technical replicates (Fig. 7C). The potential contamination of these samples by platelets, erythrocytes, and coagulation factors were assessed via Baize software. Results showed five samples exhibited detectable platelet, erythrocyte or coagulation contamination (Fig. 7D–F). Samples with contamination were excluded from downstream analysis.

We then applied eight machine learning algorithms, including logistic regression, kNeighbors, SVC, Random Forest, Gradient-Boost, MLP, AdaBoost and Extra Trees, to separate benign and malignant lung nodules (Fig. 7G). Top 30 proteins or clinical features were selected for modeling. The results showed that the Extra Tree had the highest F1 score, accuracy, precision, recall, and AUC values compared to the other models (Fig. 7G). The Extra Trees model prioritized tumor size, CA-125 and 28 proteins with known association with lung cancer (Fig. 7H), including Fascin-1 (FSCN1) (Chen et al, 2019), CD98hc (SCL3A2) (Li et al, 2023), Platelet-derived growth factor D (PDGF) (Donnem et al, 2008), CD16a (FCGR3A) (Wang et al, 2023), Hepcidin (HAMP) (Duru et al, 2012; Sonnweber et al, 2014), Azurocidin 1 (AZU1) and Collagen XV (COL15A1) (Liang et al, 2023), suggesting that our methodology successfully identified important proteins related to the disease phenotype. Further validation of these potential protein biomarkers in independent lung nodule cohorts is beyond the scope of this study.

## Discussion

Circulating blood proteomics has become an indispensable strategy for biomarker discovery, due to its minimally invasive sampling approach and the capacity to systematically characterize disease-associated molecular signatures within blood components (Cai et al, 2023; Deutsch et al, 2021; Geyer et al, 2017; Niu et al, 2022; Niu et al, 2025). MS-based plasma proteomics analysis faces challenges in proteome coverage due to the wide dynamic range of plasma protein abundances. Various technological advancements—such as the Top14 kit (Shen et al, 2020), acid precipitation (Albrecht et al, 2025; Viode et al, 2023), and recently emerged

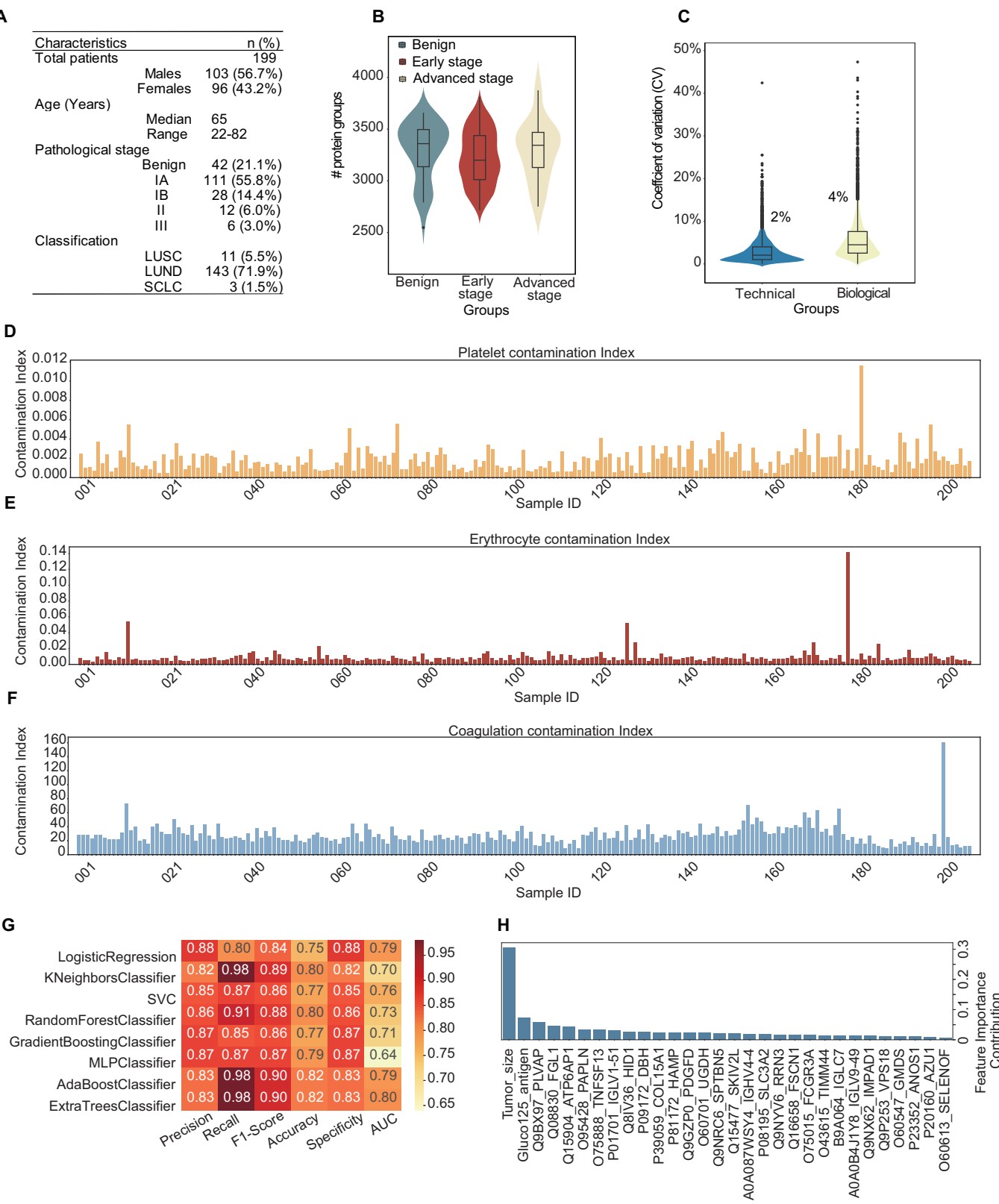

◄ **Figure 7. OmniProt-enabled plasma proteomics facilitates early detection of lung cancer.**

(A) Clinical characteristics of lung cancer cohort. (B) Global distribution of identified protein groups in the plasma proteome. (C) Reproducibility analysis in technical and biological replicates. Platelet- (D), erythrocyte- (E) and coagulation- (F) contamination profiling in the lung cancer cohort. (G) Machine learning classifier performance comparison. (H) Feature importance ranking of diagnostic biomarkers. The box in each group spans from the first to the third quartile, with a horizontal line indicating the median value. The whiskers extend from the box to represent the range of data within 1.5 times the interquartile range. Source data are available online for this figure.

nanoparticle enrichment methods (Blume et al, 2020; Liu et al, 2025; Ma et al, 2023; Wang et al, 2024)—aim to improve the depth of plasma protein identification. The nanoparticle-based approach is gaining popularity, but previous literature (Blume et al, 2020; Li et al, 2024; Ma et al, 2023) and our current study report significant variability in protein identification results (ca 2000–7000 protein groups) across plasma samples even with similar LC-MS workflows. Based on prior evidence, we hypothesized that platelet and erythrocyte contamination might contribute to this variability. In this study, we established OmniProt—a deep-coverage nanoparticle-enhanced plasma proteome preparation workflow—accompanied by the Baize computational toolkit, which enables rapid assessment of platelet, erythrocyte, and coagulation-related contamination in clinical plasma samples. This advancement carries implications for several key contributions to the plasma proteome community.

In this study, we confirm that the variability of circulating plasma protein numbers identified by nanoparticle-based methods are primarily attributed to contamination from platelet, erythrocyte and coagulation. Blood-derived contamination arises not only from sample collection, but also from multiple pre-analytical and analytical factors that influence the extent and nature of protein corona contamination in NP-based plasma proteomics. Buffer composition and nanoparticle surface chemistry further modulate the binding dynamics between plasma proteins and the nanoparticle surface. Sheibani et al (Sheibani et al, 2021) reported that high-concentration nanoparticles can promote protein flocculation and aggregation during sample preparation, potentially leading to artificial enrichment of non-target proteins. We validated this observation through cryo-EM and TEM analysis, confirming that reduced nanoparticle concentrations decrease protein flocculation and aggregation on grids. Additionally, a recently study by Korff et al (Korff et al, 2025) compared contamination susceptibility across five sample preparation methods, showing that physicochemical properties of NPs and buffer conditions affected contamination levels, with phosphate-based buffers facilitating platelet binding to non-magnetic NPs. Recently, Tang et al (Tang et al, 2025) reported a comprehensive analysis of the methodological challenges in mass spectrometry-based protein corona characterization, identifying key issues in protein corona formation, isolation, and analysis that affect data reliability, and proposed standardized protocols to improve reproducibility and accuracy in nanomedicine applications. Collectively, these factors demonstrate that achieving reliable and reproducible results in nanoparticle-based plasma proteomics requires careful consideration of the entire analytical workflow, from sample preparation to data interpretation.

This study presents a systematic approach to evaluate and quantify contamination effects in nanoparticle-based plasma proteomics. In our spike-in experiments, the protein numbers identified from plasma increase sharply when the contamination of platelet and erythrocyte increases. Although such observations have been reported for neat plasma by Geyer et al (Geyer et al, 2019), no study has comprehensively investigated the impact of contamination in nanoparticle-based circulating plasma proteomics.

After realizing the impact of blood contamination in plasma protein identification, We clarify that such contamination influence protein quantification. Our data showed that the abundances of circulating plasma proteins in the PPP samples exhibited weak correlation (r = 0.54–0.73) with those in plasma samples with various degrees of platelet contamination.

We further developed a software tool called Baize to evaluate contamination in nanoparticle-based circulating plasma proteomics experiments. We discovered and validated protein biomarkers for evaluating contaminations of platelet, erythrocyte, and coagulation-related proteins. A Spline regression algorithm was subsequently implemented to assess contamination severity and quantify contamination levels. The evaluation is implemented in an open-access web-based software tool named Baize (https://www.guomics.com/software/Baize).

We also established a silica nanoparticle-based sample preparation workflow for enriching low-abundance plasma proteins. In previous studies, multiple nanoparticle enrichment methods have been used for plasma proteomics (Blume et al, 2020; Liu et al, 2025; Ma et al, 2023; Wang et al, 2024). Representative approaches include magnetic material-mediated low-abundance protein enrichment (5–10 types) as commercialized by Seer (Blume et al, 2020; Ferdosi et al, 2022; Suhre et al, 2024), molecular sieve-based methods reported by Ma et al (Ma et al, 2023) and Li et al (Li et al, 2024), and silica-based strategies (ProteoFish) developed by Tan's team (Liu et al, 2025; Wang et al, 2024). During our preliminary experiments, molecular sieve-type nanoparticles exhibited performance degradation in low-abundance protein enrichment following moisture absorption, with unrecoverable efficiency even after secondary drying—indicating potential risks for cross-laboratory and longitudinal large-cohort applications (Sun et al, 2020). Silica-based NPs show a high degree of stability, batch-to-batch consistency, and cost-effectiveness. Tan's team achieved an average of ~1174 protein group identifications using silica nanoparticle enrichment on an Exploris 480 instrument with a 120-min LC gradient method (Wang et al, 2024). In this study, we systematically compared the effects of different types (morphologies, sizes) of silica nanoparticles on plasma proteome identification and selected NP23 (solid, 500 nm) with the highest identification capacity and stability for subsequent analyses. We then optimized key parameters in protein corona formation, including plasma diluent, incubation time, washing steps, and thermal stability, establishing an optimized workflow named OmniProt. OmniProt enables the identification of ca 3000–6000 proteins across different plasma samples using Astral with 60 SPD throughput, yielding 1.7- and 4.8-fold more protein groups and 2.6- and 5.7-fold more peptides compared to the Top14-protein depleted samples and the neat plasma samples, respectively. In the mixed-species benchmarking experiment, the OmniProt method exhibited high precision and accuracy comparable to neat samples. Nevertheless, the silica beads

require multiple centrifugation steps, which introduce difficulties for automated sample preparation. Increasingly, magnetic bead-based nanoparticles have been developed for plasma proteomics (Blume et al, 2020; Ferdosi et al, 2022). A magnetic version of OmniProt should be developed in future studies.

Our study also presents a comprehensive Astral spectral library for plasma proteomics. Several spectral libraries for the plasma proteome have been published. Wang et al (Wang et al, 2024) established a $SiO_2$ nanoparticle-derived library covering 2564 proteins, whereas Ma et al (Ma et al, 2023) developed a Zeolite NaY nanoparticle-based database containing 6524 proteins. In this study, we constructed a comprehensive spectral library for DIA data analysis using 21 chemically distinct nanoparticles combined with peptide fractionation, containing 126,661 peptide precursors corresponding to 10,109 protein groups. This spectral library was established using cell-free plasma samples devoid of cellular contamination and represents a publicly available, extensively characterized plasma proteome spectral library for community use.

Finally, we validate the OmniProt and Baize tools with a patient cohort of 193 individuals with CT-indistinct benign small pulmonary nodules or early-stage lung cancers. We identified on average ~4000 proteins per plasma sample with OmniProt using Astral at 60 SPD throughput, with five samples flagged for platelet/erythrocyte or coagulation contamination. Furthermore, by integrating proteomic data with clinical information using machine learning, we developed a blood-based classifier to distinguish benign pulmonary nodules from early-stage lung cancers, achieving an AUC of 0.8. This result demonstrates that OmniProt provides a robust pipeline for the development of multiplexed protein biomarker detection in lung cancer screening.

While Baize effectively identifies plasma samples with potential platelet, erythrocyte or coagulation-related contamination, our study has several limitations. First, our OmniProt workflow has shown excellent performance in clinical cohorts, but the use of non-magnetic nanoparticles necessitates an additional centrifugation step, and we acknowledge that magnetic nanoparticles would offer improved workflow efficiency for clinical implementation—a direction we are pursuing in our ongoing research. Additionally, our study primarily compared OmniProt against neat plasma and the Top14 kit methods due to limited availability of reagents from other published nanoparticle-based studies and prohibitive costs of commercial platforms for comprehensive comparative analysis. Fortunately, a recent preprint study by Korff et al (Korff et al, 2025) provides a comprehensive comparison of OmniProt with several alternative methods, including additional nanoparticle-based approaches and perchloric acid-assisted precipitation, offering valuable complementary data that addresses this limitation. Second, Baize was developed using 6–10 groups of plasma samples with varying contamination levels; due to limited data size, the current implementation can only assess contamination severity but cannot systematically correct for contamination-induced quantitative biases in proteomic analyses. Notably, Korff et al (Korff et al, 2025) reported that high-speed centrifugation can partially mitigate platelet contamination effects, offering a potential alternative to sample exclusion. Future work incorporating large-scale spike-in standards and modeling across laboratories may enable contamination-aware normalization and correction. Although OmniProt demonstrated high accuracy in distinguishing small malignant nodules from benign ones in our clinical cohort, the data from the retrospective study necessitate future validation in prospective cohorts.

In conclusion, we systematically investigated contamination in nanoparticle-based circulating plasma proteomics, and found that contamination of platelet, erythrocyte, and coagulation-related proteins substantially altered protein identification and quantification. We then identified protein biomarkers to evaluate these contamination and present an open-access software Baize for evaluating the contamination of NP-based plasma proteomics. Together with optimized sample preparation and standardized DIA-MS workflows, our study provides a comprehensive and reproducible framework for deep plasma proteomics with built-in contamination control.

# Methods

### Reagents and tools table

| Reagent/Resource | Reference or Source | Identifier or Catalog Number |
|---|---|---|
| **Experimental models** | | |
| Plasma, serum, platelet, erythrocyte samples | The First Affiliated hospital, Zhejiang University, school of medicine | N/A |
| **Chemicals, enzymes and other reagents** | | |
| Phosphate-buffered saline | SIGMA ALDRICH | P3813 |
| 3-[(3-cholamidopropyl) dimethylammonio]-1-propanesulfonate | GPCSCI | AP756 |
| Ammonium bicarbonate | GENERAL REAGENT | A6141 |
| Ethylenediaminetetraacetic acid | SIGMA ALDRICH | 798681 |
| Urea | SIGMA ALDRICH | 56180 |
| Tris(2-carboxyethyl) phosphine | SIGMA ALDRICH | C4706 |
| Thiourea | SIGMA ALDRICH | 200-543-5 |
| Iodoacetamide | SIGMA ALDRICH | 205-630-1 |
| Ammonium bicarbonate | Shanghai Titan Scientific | 016310809 |
| Trifluoroacetic acid | Fisher Scientific | PI28904 |
| Trypsin | Beijing Life Proteomic | HLS TRY001C |
| High Select Depletion Spin Columns | Thermo Fisher Scientific | A36369 |
| 3 kDa MWCO Ultra Centrifugal Filter | Thermo Fisher Scientific | 89868 |
| SOLAμ HRP 2 mg/1 ml 96-well plate | Thermo Fisher Scientific | 60209-001 |
| Acetonitrile | Fisher Scientific | 042311.K7 |
| Formic acid | Fisher Scientific | 28905 |
| Ammonia solution/ Ammonium hydroxide | Aladdin | A112085 |
| Methanol | Fisher Chemical | 022909.K2 |
| High Select Depletion Resin | Thermo Fisher Scientific | A36372 |
| EDTA K2 collection tubes | Hengshui Shengcixing Company | 210113 |

| Reagent/Resource | Reference or Source | Identifier or Catalog Number |
|---|---|---|
| Serum (red-top) tubes | Hengshui Shengcixing Company | 200704 |
| Lithium heparin | Gongdong Medical | GD030LH |
| Nanoparticle, NP03 | Sepax | 280660950 |
| Nanoparticle, NP10 | Baseline | 6-1-0030 |
| Nanoparticle, NP12 | Baseline | 6-2-0050 |
| Nanoparticle, NP15 | Gift from WestlakeOmics | N/A |
| Nanoparticle, NP16 | Gift from WestlakeOmics | N/A |
| Nanoparticle, NP17 | Gift from WestlakeOmics | N/A |
| Nanoparticle, NP20 | Gift from WestlakeOmics | N/A |
| Nanoparticle, NP23 | WestlakeOmics | OM2401 |
| Nanoparticle, NP24 | Gift from WestlakeOmics | N/A |
| Nanoparticle, NP30 | J&K Scientific | 9003-53-6 |
| Nanoparticle, NP34 | J&K Scientific | 60676-86-0 |
| Nanoparticle, NP40 | Gift from Dalian University of Technology | N/A |
| Nanoparticle, NP41 | Gift from Dalian University of Technology | N/A |
| Nanoparticle, NP42 | Gift from Dalian University of Technology | N/A |
| Nanoparticle, NP43 | Gift from Dalian University of Technology | N/A |
| Nanoparticle, NP45 | Agela | PA5006 |
| Nanoparticle, NP48 | Agela | JXA5003 |
| Nanoparticle, NP74 | Gift from Dalian University of Technology | N/A |
| Nanoparticle, NP82 | Gift from WestlakeOmics | N/A |
| Nanoparticle, NP85 | XFNANO | 7631-86-9 |
| Nanoparticle, NP95 | XFNANO | 9001-03-0 |
| **Software** | | |
| Code for data analysis, modeling, and figure generation | This work | GitHub: https://github.com/guomics-lab/Baize |
| Python | https://python.org | Version 3.12.5 |
| Python | https://python.org | Version 3.10 |
| DIA-NN | https://github.com/vdemichev/DiaNN | Version 1.8.1 |
| Adobe Illustrator | Adobe Systems | Version 2022 |
| Mass spectrometry data | This paper | PRIDE partner repository with the dataset identifier PXD068107 |

| Reagent/Resource | Reference or Source | Identifier or Catalog Number |
|---|---|---|
| **Other** | | |
| Vanquis Neo UHPLC System | Thermo Fisher Scientific | VN-S10-A-01 |
| Orbitrap Astral Mass Spectrometer | Thermo Fisher Scientific | BRE725660 |
| High resolution field emission scanning electron microscope | Zeiss | Crossbeam 550 |
| Ultra-High Resolution Field Emission Scanning Electron Microscopy | CIQTEK | SEM5000X |
| High Vacuum Sputter Coater | Leica | EM ACE600 |
| Thermionic (scanning) transmission electron microscope | Thermo Fisher Scientific | Talos L120C |
| Cryo-TEM | Thermo Fisher Scientific | Glacios 2 |

## Clinical sample collection

Blood samples were collected from patients with lung cancer or pulmonary benign nodules using standardized vacutainer tubes at the First Affiliated Hospital of Zhejiang University School of Medicine (Hangzhou, China), with ethical approval granted by both the Institutional Ethics Committee of the First Affiliated Hospital (Approval No. 2024-0511) and the Ethics Committee of Westlake University (Approval No. 20231218GTN001). Informed consent was obtained from all human subjects and confirmed that the experiments conformed to the principles set out in the WMA Declaration of Helsinki and the Department of Health and Human Services Belmont Report. Four types of blood collection tubes, including EDTA K2 (purple-top) and serum (red-top) tubes from Hengshui Shengcixing Company (Hebei, China), as well as Lithium Heparin (green-top) and Sodium Citrate (light blue-top) tubes from Gongdong Medical (Taizhou, China), were used to collect plasma and serum samples. All blood samples were immediately stored at 4 °C following collection, with plasma separation completed within 4 h of blood draw. Whole blood samples collected for quality marker panel establishment were first analyzed for cellular components within 4 h post-collection using an automated hematology analyzer (Shenzhen Mandray Biomedical Electronics Co., Ltd., China).

## Graphics

Figure 1A, Fig. 2A, Fig. 4A, Fig. 4F, Fig. 4I, Fig. 5A, Fig. 5F, and synopsis graphics were created with Biorender.com.

## Characterization of diverse SiO$_2$-based NPs

The morphology of SiO$_2$ NPs was analyzed using a Zeiss Gemini 550 Crossbeam (Oberkochen, Germany) and a CIQTEK SEM5000X FESEM Microscope (Anhui, China) with gold-coated copper TEM grids. For SEM sample preparation, both NPs and NP-protein coronas were first fixed overnight at 4 °C in a fixative solution (pH 7.2)

containing 2% paraformaldehyde (PFA) and 2.5% glutaraldehyde. After fixation, the samples were washed three times with 0.1 M PBS at 4 °C on a shaker for 15 min. Subsequently, the samples were rinsed three times with deionized water at room temperature on a shaker, followed by centrifugation at $7000 \times g$ for 3 min. After the final centrifugation, 100 µL of the supernatant was retained. The pellet and the 100 µL supernatant were then sonicated for 3 min in an ice bath to yield a suspension. For SEM imaging, 2.5 µL of the suspension was dropped onto a silicon wafer and dried for 2 min using a drying system (Leica, Germany). The dried samples were then coated with an 8 nm layer of platinum using an EM ACE600 High Vacuum Sputter Coater (Leica, Germany). The platinum-coated samples were subsequently used for further analysis. To assess potential contamination from protein aggregates across different nanoparticle concentrations, both room temperature transmission electron microscopy (RT-TEM) and cryo-transmission electron microscopy (cryo-TEM) were employed. Two nanoparticle concentration conditions were tested: Condition 1 used 0.5 mg nanoparticles in 180 µL protein corona reaction system (concentration: 2.78 mg/mL), representing the standard OmniProt protocol concentration, while Condition 2 employed 0.5 mg nanoparticles in a 90 µL protein corona reaction system (concentration: 5.55 mg/mL). After protein corona formation, samples were washed three times with buffer 3 as described above, and then resuspended in 100 µL water. For electron microscopy analysis, 2.5 µL of the suspension was deposited onto hydrophilic TEM grids (treated with Coolglow glow discharge, SuPro Instruments) and analyzed using a Talos L120C (RT-TEM) and Glacios 2 (cryo-TEM) from Thermo Fisher Scientific.

## Clinical sample pre-treatment

In the platelet contamination assessment experiment, we obtained four types of plasma samples from whole blood using four isolation protocols. The first protocol involved two centrifugation cycles at $2000 \times g$ for 15 min each, with the plasma supernatant collected after the second centrifugation. The second protocol used a single centrifugation step at $2000 \times g$ for 15 min, after which the plasma supernatant was collected. The third protocol utilized a lower centrifugal force of $200 \times g$ for 15 min, collecting the plasma supernatant. The fourth protocol employed overnight sedimentation before supernatant collection. All processed plasma samples were immediately aliquoted into 1.5 mL tubes and stored at −80 °C. Prior to experimental use, frozen aliquots were thawed at 4 °C under controlled conditions.

Purified PRP, PPP, erythrocyte, and platelet samples were obtained through the following procedures. The blood was first centrifuged at $200 \times g$ for 10 min, retaining both the pellet and supernatant. To minimize contamination from the supernatant, approximately 0.5 cm of supernatant adjacent to the erythrocyte layer was discarded, resulting in PRP samples. The pellet was centrifuged at $2000 \times g$ for 15 min, and the upper layer (containing residual plasma, buffy coat, and ~1 mL erythrocytes) was removed, yielding erythrocyte-enriched fractions. Subsequently, erythrocytes were washed twice with 6 mL of phosphate-buffered saline (PBS) containing 1.6 mg/mL EDTA, and centrifuged at $2000 \times g$ for 15 min (discarding the supernatant and ~1 mL erythrocytes each time) to isolate purified erythrocytes. The supernatant from the first centrifugation underwent a second centrifugation at $200 \times g$ for 10 min to harvest purified PRP samples. This PRP supernatant was centrifuged sequentially at $200 \times g$ for 10 min (the pellet was discarded), followed by two centrifugations at $2000 \times g$ for 15 min, and

the upper layer collected as PPP samples. Platelet pellets from the $2000 \times g$ spins were washed twice with 4 mL of EDTA-PBS and centrifuged at $2000 \times g$ for 15 min, and then resuspended in PBS buffer. Purified PRP, PPP, erythrocytes, and platelets were quantified using an automated hematology analyzer (Shenzhen Mandray Biomedical Electronics Co., Ltd., China).

## Proteome sample preparation

For plasma samples without protein enrichment (neat), 2 µL of plasma was processed using a standard in-solution digestion protocol (Cai et al, 2023). All chemical reagents, unless otherwise specified, were obtained from Sigma-Aldrich (St. Louis, MO, USA). Specifically, 50 µL of lysis buffer containing 8 M urea and 2 M thiourea, dissolved in 100 mM ammonium bicarbonate (ABB, Shanghai Titanchem, Catalog #01005385, Shanghai, China), was added for protein denaturation. The mixture was then treated with a final concentration of 10 mM Tris(2-carboxyethyl)phosphine (TCEP, Sigma, Catalog #T4708) and 40 mM iodoacetamide (IAA, Sigma, Catalog #SLCD4031) in the dark for 30 min to facilitate reduction and alkylation. Subsequently, a 100 mM ABB dilution solution was used to reduce the urea concentration to below 1.2 M, followed by the addition of 1 µg of trypsin (Hualishi Scientific, Beijing, China) at a 1:50 ratio for overnight digestion. Finally, 30 µL of 10% Trifluoroacetic acid (TFA, Fisher Scientific, Loughborough, UK) is added to terminate the digestion. The resulting tryptic peptides are purified using a SOLAµ HRP 2 mg/1 ml 96-well plate (Thermo Fisher Scientific, Germany) and dried under vacuum. Before MS analysis, peptides were dissolved in 2% MS-grade ACN with 0.1% (v/v) formic acid. The peptide amounts were quantified using ScanDrop2 (Analytik Jena, Germany). The OmniProt-based sample preparation began with 15 µL of plasma samples, which were first diluted with 75 µL of a 1×PBS buffer containing 0.05% 3-[(3-cholamidopropyl)dimethylammonio]-1-propanesulfonate (CHAPS, GPC Biotechnology GmbH, Germany) and 0.02% ammonia (referred to as buffer 2). Next, 0.5 mg of NP solution was added, and the mixture was incubated on a shaker at 30 °C and 220 rpm for 30 min. After incubation, the soft corona was washed with a 33% diluted buffer 2, using centrifugation at $7000 \times g$ for 10 min, repeated three times. Finally, the soft corona was treated using the same protein denaturation, digestion, and desalting methods as the neat plasma samples. The Top14 kit (Thermo Fisher Scientific, Germany) method was performed as previously described (Shen et al, 2020). Briefly, 10 µL of plasma samples were depleted using the Top14 kit and then concentrated to 50 µL through a 3 kDa MWCO Ultra Centrifugal Filter (Merck, Germany). Next, the remaining protein was denatured in 8 M urea at 30 °C for 30 min, reduced with 10 mM TCEP, and alkylated with 40 mM IAA. Subsequently, 75 µL of 100 mM ABB solution was added to dilute the urea concentration. Trypsin was introduced at 1.0 µg to initiate proteolytic digestion. The digestion was then stopped by adjusting the pH to 2–3 using 1% trifluoroacetic acid (Thermo Fisher Scientific). The resulting tryptic peptides were desalted via a SOLAµ HRP 2 mg/1 ml 96-well plate and stored at −80 °C for downstream analyses.

## Enrichment of plasma proteins using various NPs for building a matching library

We utilized pooled plasma samples from patients (Dataset EV2) with lung cancer to enrich low-abundance proteins using various

NPs, following the OmniProt method. Approximately 200 μg of plasma digests were loaded onto a Waters XBridge column (2.1 mm × 150 mm, BEH C18, 5 μm) with a Thermo Scientific UltiMate™ 3000 RSLC LC system equipped for basic-pH reverse phase liquid chromatography (basic pH RPLC), producing 120 fractions that were subsequently combined into either 30 or 60 fractions. These were evaporated and resuspended for further analysis. Each fraction was loaded into a trap column and separated on a custom-made analytical column (75 μm i.d. × 15 cm, 1.9 μm) using a 24-min LC-MS method with a Vanquish™ Neo UHPLC system (Thermo Fisher Scientific, Germany). The initial gradient began at 8% buffer B (buffer B: 80% MS-grade ACN with 0.1% (v/v) formic acid; buffer A: 2% MS-grade ACN with 0.1% (v/v) formic acid), increasing to 10% B over 1.5 min, then to 30% B over 16 min, and finally to 40% B over 2 min, with 4.3 min reserved for column cleaning and equilibration. The eluted peptides were analyzed using an Orbitrap Astral mass spectrometer (Thermo Fisher Scientific, Germany), with FAIMS voltage set at −42 V, a full scan resolution of 240,000, and a scan range of 380–980 Th. MS/MS scans were conducted within the same range, employing a DIA isolation window of 1 Da.

## DIA acquisition

First, approximately 300 ng peptides were loaded into a trap column and subsequently separated on an analytical column (75 μm i.d. × 15 cm, 1.9 μm, custom-made) using a 24-min LC-MS method. The initial liquid phase gradient for LC separation commenced with 8% buffer (buffer B: 80% ACN with 0.1% (v/v) formic acid; buffer A: 2% MS-grade ACN with 0.1% (v/v) formic acid), increasing to 10% B over 1.5 min, then to 30% B over 16 min, and finally to 40% B over 2 min. The LC method incorporated a 4.3-min column cleaning and equilibration phase between runs. The mass spectrometry parameters for DIA acquisition were consistent with those used for library construction, except that the isolation window was set to 2 Da.

## DIA-MS analysis

DIA files were analyzed with DIA-NN (v1.8.1) (Demichev et al, 2020). For platelet-, erythrocyte-, and coagulation-related DIA files, analyses were conducted against the reviewed UniProt Human database (20,377 entries, downloaded in May 2020) with default parameters and Match-Between-Runs (MBR) enabled. For all other DIA datasets, a custom spectral library comprising 126,661 peptide precursors mapped to 10,109 protein groups (generated in this study) was employed, using identical default settings and MBR. A 1% false discovery rate (FDR) threshold was applied at both peptide precursor and protein levels, with quantification performed in Robust LC mode to ensure analytical precision. DIA files generated from the human and bovine plasma benchmarking experiment were analyzed using DIA-NN (v1.8.1) against a combined database comprising the reviewed UniProt Human database (20,377 entries, downloaded in May 2020) and the reviewed UniProt Bovine database (6048 entries, downloaded in January 2025), with default parameters and MBR.

## Quantification and statistical analysis

### Bioinformatics analysis

Statistical analysis of the data was performed using R software (v4.1.2) and Python (v3.12.5 and v3.10), which included the use of heatmap and R package plot functions. The proteins in the heatmaps were hierarchically clustered using the centroid method. The main report containing peptide and protein quantification results was retained for downstream evaluation. OmniProt workflow precision was assessed by calculating the CV for each protein across biological replicates. We used bovine-specific peptide precursors to evaluate the accuracy of the OmniProt workflow. Peptide precursor intensities were log-transformed, and technical variations across replicates were corrected through global median normalization of log-intensities. Then we compared measured bovine-specific peptide precursors' fold changes of spiked-in ratios with their expected theoretical values.

To identify platelet- and erythrocyte-related biomarkers, proteins with missing values in more than 50% of samples were first excluded. Candidate markers for platelet or erythrocyte contamination were selected from the top 100 proteins most strongly correlated with platelet concentration changes. Among these, the 30 most abundant were prioritized as potential biomarkers. Candidate markers for coagulation-related contamination were identified through the following workflow. First, proteins with missing values in more than 50% of samples were removed. The abundance of each protein under each condition was calculated as the mean of three biological replicates. Next, proteins meeting an adjusted p-value < 0.05 and fold change >5 between plasma and serum samples were selected. From this subset, the top 20 most abundant proteins were prioritized as potential coagulant contamination markers.

### Development of Baize software

The Baize algorithm operates through a unified computational workflow designed for automated contamination assessment. Users

**The paper explained**

**Problem**
Plasma proteomics represents a powerful approach for discovering circulating protein biomarkers to monitor health and disease. Nanoparticle-based strategies have been employed to enhance the detection of low-abundance proteins via protein corona formation. However, studies utilizing this approach exhibit considerable variability in reported protein identification numbers, highlighting the importance of uncovering the sources of the variability.

**Results**
We developed a robust and standardized nanoparticle-based proteomic method, named OmniProt. Through systematic experimental design, we identified that plasma samples contaminated by platelets or erythrocytes artificially elevates protein identifications and impairs quantification accuracy. We further conducted a series of assays to define specific protein markers for three types of contamination: erythrocytes, platelets, and coagulation-related proteins. Finally, we integrated these findings into Baize, an open-source software tool that automatically evaluates data quality, detects contamination signals based on the identified markers, and flags affected samples for further review.

**Impact**
Contaminated plasma leads to inflated protein identifications. Therefore, it is essential to screen for blood contamination in plasma proteomic analyses. The open-source software tool, Baize, enables effective assessment of this contamination.

input a protein matrix (generated by search engines such as DIA-NN), and the software calculates contamination indexes for platelets, erythrocytes, and coagulation-related proteins using predefined biomarker panels (e.g., PF4 for platelets, HBA1 for erythrocytes, FGA for coagulation). These indexes are derived by normalizing the summed intensities of contamination-specific biomarkers against the total plasma proteome intensity. The output provides each sample's contamination profiles, enabling rapid quality control decisions without requiring manual intervention.

## Data availability

The preprocessed data have been deposited in the PRIDE (Perez-Riverol et al, 2022) under dataset identifier PXD068107. Additionally, all original code supporting this study is available in the GitHub repository at https://github.com/guomics-lab/Baize.

The source data of this paper are collected in the following database record: biostudies:S-SCDT-10_1038-S44321-025-00346-9.

## Peer review information

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

## Acknowledgements

This work is supported by grants from the Noncommunicable Chronic Diseases-National Science and Technology Major Project (2024ZD0533300), the "Pioneer" and "Leading Goose" R&D Program of Zhejiang (2023C03056, 2024SSYS0035), National Key R&D Program of China (Grant No. 2022YFF0608403 to Dr Yi Zhu), National Natural Science Foundation of China (82303849 to Dr Yuanqi Liu). The language of the first draft has been polished with Claude Sonnet 4 before proofreading by the co-authors.

## Author contributions

**Huanhuan Gao**: Conceptualization; Resources; Data curation; Software; Formal analysis; Supervision; Validation; Investigation; Visualization; Methodology; Writing—original draft; Project administration; Writing—review and editing. **Yuecheng Zhan**: Data curation; Methodology. **Yuanqi Liu**: Resources. **Zhiyi Zhu**: Formal analysis; Visualization. **Yuxiu Zheng**: Methodology. **Liqin Qian**: Methodology. **Zhangzhi Xue**: Formal analysis; Visualization. **Honghan Cheng**: Formal analysis; Visualization. **Zongxiang Nie**: Software. **Weigang Ge**: Resources. **Senlin Ruan**: Resources. **Jiaxu Liu**: Resources. **Jikai Zhang**: Resources. **Yingying Sun**: Writing—review and editing. **Lei Zhou**: Resources; Methodology. **Dongyue Xun**: Methodology. **Yingrui Wang**: Writing—review and editing. **Heyun Xu**: Resources; Supervision. **Huiwen Miao**: Resources. **Yi Zhu**: Supervision. **Tiannan Guo**: Conceptualization; Supervision; Funding acquisition; Investigation; Methodology; Writing—original draft; Project administration; Writing—review and editing.

Source data underlying figure panels in this paper may have individual authorship assigned. Where available, figure panel/source data authorship is listed in the following database record: biostudies:S-SCDT-10_1038-S44321-025-00346-9.

## Disclosure and competing interests statement

TG and YZ are shareholders of Westlake Omics (Hangzhou) Biotechnology Co., Ltd., where OmniProt and related technologies are commercialized. Two patents related to Baize technologies have been filed, with the application numbers CN 202510492870.3 and CN 202510492911.9. ZZ, WG, YC, and YZ are employed by Westlake Omics (Hangzhou) Biotechnology Co., Ltd. The remaining authors declare no competing interests.

