## [Peer Review File · EMBO Molecular Medicine]

Systematic evaluation of blood contamination in nanoparticle-based plasma proteomics

Huanhuan Gao, Yuecheng Zhan, Yuanqi Liu, Zhiyi Zhu, Yuxiu Zheng, Liqin Qian, Zhangzhi Xue, Honghan Cheng, Zongxiang Nie, Weigang Ge, Senlin Ruan, Jiaxu Liu, Jikai Zhang, Yingying Sun, Lei Zhou, Dongyue Xun, Yingrui Wang, Heyun Xu, Huiwen Miao, Yi Zhu, and Tiannan Guo

Corresponding author: Tiannan Guo (guotiannan@westlake.edu.cn)

Review Timeline:

Submission Date:	10th Nov 25
Editorial Decision:	11th Nov 25
Revision Received:	12th Nov 25
Accepted:	13th Nov 25

Editor: Jingyi Hou

Transaction Report:

Please note that the manuscript was previously reviewed at another journal and the reports were taken into account in the decision making process at EMBO Molecular Medicine. Since the original reviews are not subject to EMBO Press' transparent review process policy, the reports and author response cannot be published.

11th Nov 2025

Dear Tiannan,

Thank you for submitting your manuscript to EMBO Molecular Medicine. We have carefully reviewed the revised version along with your point-by-point response to the reviewers' comments from the other journal. It is clear that you have thoroughly addressed all the issues raised, and the reviewers were satisfied with your revisions. Overall, we believe the study makes an important contribution to the field, and we are pleased to inform you that it will be accepted for publication pending minor editorial revisions, as outlined below.

1. Please provide up to five keywords in the manuscript file.
2. "DECLARATION OF INTERESTS" should be renamed to "DISCLOSURE AND COMPETING INTERESTS STATEMENT".
3. Remove "Authors' contribution" section from the manuscript file.
4. In Methods, for human samples, please include a statement that informed consent was obtained from all human subjects and that the experiments conformed to the principles set out in the WMA Declaration of Helsinki and the Department of Health and Human Services Belmont Report.
5. Appendix:
 - Rename the file "Supplementary Information" to "Appendix" and update all related figure labels accordingly (e.g., Appendix Figure S1, Appendix Figure S2, etc.). The callouts in the manuscript file need to be updated as well.
 - Add a Table of Contents on the first page, including corresponding page numbers for the listed items.
 - Remove all line numbers from the final version of the file.
6. Rename "Appendix Tables 1, 2, 3, and 5" to "Dataset EV1-EV4" and include a legend in the corresponding Excel file, providing the dataset name and a brief description of its contents in a separate tab or worksheet. Rename "Appendix Table 4" to "Table EV1" and add a legend at the top of the table.
7. Every published paper now includes a 'Synopsis' to further enhance discoverability. Synopses are displayed on the journal webpage and are freely accessible to all readers. They include a short stand first (maximum of 300 characters, including space) as well as 2-5 one-sentence bullet points that summarize the paper. Please write the bullet points to summarize the key NEW findings. They should be designed to be complementary to the abstract - i.e. not repeat the same text. We encourage inclusion of key acronyms and quantitative information (maximum of 30 words / bullet point). Please use the passive voice. Please attach these in a separate file or send them by email, we will incorporate them accordingly.

Please provide visual abstract to illustrate your article as a PNG file 550 px wide x 300-600 px high.

8. Please provide 'The paper explained': EMBO Molecular Medicine articles are accompanied by a summary of the articles to emphasize the major findings in the paper and their medical implications for the non-specialist reader. Please provide a draft summary of your article highlighting

9. Please download and fill our Reagents and Tools Table template (.docx), which you can find in our author guidelines: <https://www.embopress.org/page/journal/17574684/authorguide#structuredmethods>. When submitting your revised manuscript, please DO NOT include the Reagents and Tools Table in the Methods section of the manuscript but upload it as a separate file choosing the file type "Reagent Table".

10. Please upload a complete author checklist, which you can download from our author guidelines (<https://www.embopress.org/page/journal/17574684/authorguide#submissionofrevisions>). Please insert information in the checklist that is also reflected in the manuscript.

11. At EMBO Press we ask authors to provide source data for the main manuscript figures. You have already received a separate email with instructions for providing source data with your revised manuscript, including how to upload and organize the files.

12. The references need to be formatted according to the EMBO Molecular Medicine reference style

- Citations should be listed in alphabetical order.
- Please list up to 10 co-authors of a paper before adding et al. in the reference list.
- Remove DOI for published papers.

13. Data availability

- Remove the information under "lead contact". Merge the remaining information under "Resource Availability" and use the heading "Data Availability" instead of "Resource Availability". Remove the subheadings "Materials Availability" and "Data and Code availability". You may refer to any published paper for formatting examples.
- Please provide specific URL for PXD068107 in the data availability statement and make sure they will be made publicly available upon acceptance of the manuscript.

14. Remove information about BioRender from the "Acknowledgments" and move it to a dedicated "Graphics" section in the "Methods" using this format:

Graphics:

(some of the... OR Figure #... OR synopsis) Graphics were created with BioRender.com.

15. Please address the following issues in figure legends:

- Please indicate the statistical test used for data analysis in the legends of figures 6F, G
- Please note that the box plots need to be defined in terms of minima, maxima, centre, bounds of box and whiskers, and percentile in the legends of figures 3E, 4G, J; 5G, 7B, C.

16. Please correct the order and the headings of the manuscript sections to: Abstract / Keywords / The Paper Explained / Introduction / Results / Discussion / Methods / Data Availability / Acknowledgements / Disclosure and Competing

- Interests Statement / References / Figure Legends

We look forward to seeing a revised form of your manuscript as soon as possible.

Sincerely,
Jingyi

Jingyi Hou
Senior Editor
EMBO Molecular Medicine

To submit your manuscript, please follow this link:

<https://embomolmed.msubmit.net/cgi-bin/main.plex>

The authors addressed the remaining formatting issues.

13th Nov 2025

Dear Tiannan,

Thank you for sending us the revised manuscript. We are pleased to inform you that your manuscript is accepted for publication and is now being sent to our publisher to be included in the next available issue of EMBO Molecular Medicine.

Sincerely,
Jingyi

Jingyi Hou
Senior Editor
EMBO Molecular Medicine
